
# A multi-sourced assessment of the spatio-temporal dynamic of soil saturation in the MARINE flash flood model

Judith Eeckman[1], Hélène Roux[1], Audrey Douinot[2], Bertrand Bonan[3], and Clément Albergel[3,4]

[1]Institut de Mécanique des Fluides de Toulouse (IMFT), Université de Toulouse, CNRS - Toulouse, FRANCE
[2]Luxembourg Institute of Science and technology, ERIN, Luxembourg
[3]CNRM, Université de Toulouse, Météo-France, CNRS, Toulouse, France
[4]now at European Space Agency Climate Office, ECSAT, Harwell Campus, Didcot, Oxfordshire, UK

**Correspondence:** Eeckman Judith ju.eeckman@gmail.com

**Abstract.** The MARINE hydrological model is a distributed model dedicated to flash flood simulation. Recent developments of the MARINE model are exploited in this work: on the one hand, formerly relying on water height, transfers of water through the subsurface now take place in a homogeneous soil column based on the volumetric soil water content (SSF model). On the other hand, the soil column is divided into two layers, which represent respectively the upper soil layer and the deep weathered rocks (SSF-DWF model). The aim of the present work is to assess the performances of these new representations for the simulation of soil saturation during flash flood events. An exploration of the various products available in the literature for soil moisture estimation is performed. The performances of the models are estimated with respect to several soil moisture products, either at the local scale or spatially extended: i) The gridded soil moisture product provided by the operational modeling chain SAFRAN-ISBA-MODCOU; ii) The gridded soil moisture product provided by the LDAS-Monde assimilation chain, based on the ISBA-a-gs land surface model and assimilating satellite derived data; iii) the upper soil moisture hourly measurements taken from the SMOSMANIA observation network; iv) The Soil Water Index provided by the Copernicus Global Land Service (CGLS), derived from Sentinel1/C-band SAR and ASCAT satellite data. The case study is performed over two French Mediterranean catchments impacted by flash flood events over the 2017-2019 period. The local comparison of the MARINE outputs with the SMOSMANIA measurements, as well as the comparison at the basin scale of the MARINE outputs with the gridded LDAS-Monde and CGLS data lead to the same conclusions: both the dynamics and the amplitudes of the soil moisture simulated with the SSF and SSF-DWF models are better correlated with both the SMOSMANIA measurements and the LDAS-Monde data than the outputs of the base model. The opportunity of improving the two-layers model calibration is then discussed. In conclusion, the developments presented for the representation of subsurface flow in the MARINE model enhance the soil moisture simulation during flash floods, with respect to both gridded data and local soil moisture measurements.

## 1 Introduction

The risk associated with flash flood events is of growing importance, in particular in the Mediterranean area (Payrastre et al., 2011; Ruin et al., 2014; Suárez-Almiñana et al., 2019). Since extreme precipitation events are expected, with good confidence, to increase both in frequency and in amplitude in the context of a changing climate (IPCC, 2014), the performances of the



modeling tools available for operational purposes are of increasing stake. The main variable of interest for flood simulations at
the catchment scale is usually the integrative discharge variable. However, surface runoff, itself controlled by soil infiltration
rates, is shown to exacerbate both human and material risks during extreme events (Vincendon et al., 2010). The representation
of soil processes in the models is thus a key factor for flash flood simulation (Berthet et al., 2009).

Among the variety of models developed for flash flood simulation, a large panel of formalism is applied to model the sub-
surface, from no consideration of infiltration flows (Berthet, 2010), to reservoir-like representations of the subsurface or to
detailed parametrizations of the soil physics. In reservoir-like representations, vertical flows can be parametrized through sim-
ple calibrated relations, in particular through linear relations (Perrin et al., 2003), or exponential relations. Other approaches
apply a more physically-oriented representation of infiltration in the subsurface based on the Richard's equation. In this case,
the controlling coefficients are whether calibrated (Roux et al., 2011) or extracted from pedological and geological descriptions
(Bouilloud et al., 2010; Vincendon et al., 2010; Vannier et al., 2014).

This variety of models applied for subsurface representation reveals large uncertainties for the quantification of the trans-
fers through the subsurface during flood events. Various works quantify the sensitivity of different models to the subsur-
face parametrization (Tramblay et al., 2010; Garambois et al., 2015; Douinot et al., 2017; Edouard et al., 2018; Lovat et al.,
2019).They show that the uncertainties on the processes in the subsurface have a strong impact on both the discharge and the
surface runoff simulation during the flood events. However, the validation of simulated outputs is made hazardous by both
the lack of soil and deep ground description and by the lack of underground flows measurements (Manus et al., 2009). In
this work, an exploration of the various products available in the literature for soil moisture estimation is performed. Three
main types of data can be used to estimate the performances of event-based hydrological models regarding the soil moisture: i)
local ground measurements provide locally accurate estimations of soil moisture at shallow depths. The difficulty in comparing
ground measurements to simulation outputs stands in the fact that point measurements do not provide any spatially extended
information. In particular, the SMOSMANIA network (Calvet et al., 2007; Albergel et al., 2009; Parrens et al., 2012) consists
in 21 ground point measurements in Southern France ; ii) continuous models provide gridded information over a large area and
they can provide information for different depths and different variables. However, model outputs are necessarily biased by
structural uncertainties of the model and uncertainties on model input. For example, the SAFRAN-ISBA-MODCOU modelling
chain (Habets et al., 2008) as well as the LDAS-Monde products (Albergel et al., 2017) are both based on the ISBA surface
scheme (Noilhan and Planton, 1989; Noilhan and Mahfouf, 1996), implemented in the SURFEX plateform (Masson et al.,
2013); iii) Satellite imagery provides valuable spatially extended data. However, remote sensors are able to capture only super-
ficial reflectance of surfaces. Microwave remote sensing (RS) provides a means to quantitatively describe the water content of
a shallow near-surface soil layer. However, the variable of interest for applications in short- and medium-range meteorological
modelling and hydrological studies over vegetated areas is the root-zone soil moisture (RZSM) content, which controls plant
transpiration but is not directly observable from space. Since the near-surface soil moisture (SM) is related to RZSM through
diffusion processes, assimilation algorithms may allow its retrieval. Estimation of RZSM from intermittent remotely sensed





surface SM data had focused on the assimilation of such data into land surface models. Many studies now also suggest that
constraining those LSMs using various types of earth observations, including vegetation related earth observations, may lead
to a better representation of the RZSM.

    The MARINE model (Model of Anticipation of flows and INondations for extreme Events) (Roux et al., 2011) is a distributed, physically based hydrological model. MARINE is tested by operational French flood forecasting services for flood
risk assessment. The recent developments of the MARINE model proposed by Douinot et al. (2018) lead to an improved representation of the subsurface flow. These developments enhance the degree of refinement of the soil physics described in the
model. The impacts of this representation of the subsurface on the water discharge are extensively studied by Douinot (2016).
However, their influence on the spatial dynamic of soil saturation has not yet been explored.

Thus this work aims to assess the impacts of the developments proposed by Douinot et al. (2018) to include a physically oriented soil representation in MARINE, with respect to the soil saturation dynamics during flash flood events. The performances
of the model are estimated with respect to several soil moisture products: i) The gridded soil moisture product provided by the
operational modeling chain SAFRAN-ISBA-MODCOU, available at the 8 km x 8 km spatial resolution ; ii) The gridded soil
moisture product provided by the LDAS-Monde assimilation chain, based on the ISBA-a-gs land surface model and assimi-
lating high resolution spatial remote sensing data. This work uses the version of LDAS-Monde at the 2.5 km x 2.5 km spatial
resolution ; iii) the upper soil moisture hourly measurements taken from the SMOSMANIA observation network; iv) The Soil
Water Index provided by the Copernicus Global Land Service (CGLS), available at the kilometric resolution and derived from
Sentinel1/C-band SAR and ASCAT satellite data. The comparison between the MARINE output for soil saturation dynamics
and these three sources of data is performed both at the local point measurement scale and at the catchment scale. These prod-
ucts represent valuable indicators of the spatio-temporal dynamics of soil moisture at various scales.

    In section 2, the MARINE model along with its new developments for the soil model are described, together with the two
catchments and the events put under light for this study. The soil moisture products used in this work are also presented in this
section. In section 3, the methods employed for model set up and calibration and the comparison protocol are presented. The
last section consists in the results presentation and the last part opens the discussion concerning the validation of the simulation
of the water content of the deep underground zone.





## 2 Model and data

### 2.1 The Marine flash-flood model

#### 2.1.1 Base model (BM)

The MARINE model (Roux et al., 2011) is a distributed, physically based hydrological model. MARINE consists of three main modules: first, precipitation is separated between surface runoff and infiltration using the Green and Ampt model; then the subsurface flows are represented using an approximation of the Darcy's law; finally, the overland and river fluxes are simulated using the Saint-Venant equations simplified with kinematic wave approximation. Based on sensitivity analyses of the model (Garambois, 2012), five parameters are calibrated in MARINE for the representation of the soil and the surface: the

multiplier coefficient for soil depth maps ($C_z$), the multiplier coefficient for the spatialized saturation hydraulic conductivity used in lateral flow modelling ($C_{kss}$) the multiplier coefficient for the spatialized hydraulic conductivity at saturation that is used in infiltration modelling ($C_{kga}$), and two friction coefficients for low and high-water channels.

#### 2.1.2 The subsurface flow model (SSF)

This work uses the recent developments for the representation of the infiltration into the subsurface and the new two-layer

soil model proposed by Douinot et al. (2018). These new models are integrated into PLATHYNES, the modeling platform of the French Service for Flood Forecasting (SCHAPI). In the MARINE base model, the transfers through the subsurface are a function of the water height. However, Douinot et al. (2018) shows that expressing the subsurface flows as function of the volumic soil water content of the cell instead of its water height appears to be a more appropriate choice to represent the activation of preferential paths. Thus, Douinot et al. (2018) define a new subsurface flow model (SSF) where the lateral flows

are expressed as a function of the volumic soil water content of the cell.

#### 2.1.3 The two soil layers model (SSF-DWF)

In the soil model initially implemented in MARINE (base model, see section 2.1.1), the soil is represented by a single layer. Douinot et al. (2018) proposes a version of the soil model for which two soil layers are defined: the deep water flow model (DWF). With the DWF soil model, the soil column is subdivided by two layers which represent the 'upper soil' part and the

'weathered rock' part of the soil. This subdivision involves the definition of two new flows, in addition to the lateral flow in the upper soil to represent 1) the flows between the cells and the flows towards the drainage network in the weathered rock and 2) the vertical infiltration flow, from the 'upper soil' layer to the 'weathered rock' layer. In this DWF model, the depth of the upper layer is equal to the soil depth provided by the soil data base and the deep layer has an uniform depth over the catchment. The deep layer depth is calibrated for each catchment.


The two hypotheses made for the SSF and the DWF models can be merged to create the SSF-DWF model for the subsurface flow representation in MARINE: in the SSF-DWF model, the soil column is separated into two layers. Vertical and lateral



transfers in the upper soil layer are described as a function of volumic soil moisture. In the SSF-DWF, the flows in the deep layer remains a function of the water height. The integration of the SSF-DWF model in MARINE necessarily implies the

calibration of two additional parameters: 1) the ratio between of the hydraulic conductivity at saturation for the upper soil layer and for the deep layer; 2) the uniform depth of the deep layer. Extensive descriptions of the DWF, the SSF and the SSF-DWF model's physics and parametrization are presented in Douinot et al. (2018). The above-named acronyms are consistent with the ones used by Douinot et al. (2018).

## 2.2 Studied cases

### 2.2.1 The Ardeche at Vogue and the Orbieu at Lagrasse catchments

In this work, the study case is performed over two catchments located in the South of France, particulary submitted to flash flood events: the Ardeche river at Vogue and the Orbieu river at Lagrasse. These two catchments have been selected for this study because i) numerous flash flood events have been inventoried over the last decade over these catchments (Gaume et al., 2009) and ii) one SMOSMANIA station (Calvet et al., 2007) is installed since 2006 within each of these catchments for real-

time superficial soil moisture measurements (see section 2.3.4).

Figure 1 presents the geographic situation of these two catchments. The digital elevation model (DEM) from the French Geographic Institute (IGN) at the 25-m resolution is considered in this work. The pedological information is taken from the French national institute for agronomic research (INRA) soil data base for the Ardeche and Languedoc-Roussillon regions.

The land cover information is taken from the Corine Land Cover 2006 data base (Aune-Lundberg and Strand, 2010).

The Ardeche catchment (622 $km^2$, from 193 m.a.s.l. to 1347 m.a.s.l.) is located in the Cevennes region, exposed to intense precipitation events due to the convection of humid sea air masses over the Cevennes mountain slopes. The Orbieu catchment (236 $km^2$, from 135 m.a.s.l. to 807 m.a.s.l.) is also exposed to Mediterranean extreme events, in particular with the dramatic

flood event of October 2018. The Ardeche catchment presents a mixed geology, globally with metamorphic rocks and schists on the upper part of the catchment and sedimentary plains downstream (source: www.infoterre.brgm.fr). The land cover for the Ardeche catchment is mainly mixed forest, natural grasslands and shrubs. The Orbieu catchment consist in a sedimentary area, mainly covered by arable land. Both catchments are little anthropized. The soil is 27 cm deep on average for the Ardeche catchment, with depths between 5 cm and 50 cm, and 37 cm deep on average for the Orbieu catchment, with depths between

shallow and 73 cm. The soil texture is mainly sandy-loam for the Ardeche catchment, with silt deposits downstream and it is mainly silt and silty-loam for the Orbieu catchment. Extensive geomorphological descriptions of these two catchments can be found is Adamovic et al. (2016); Douinot (2016) and Garambois et al. (2016).





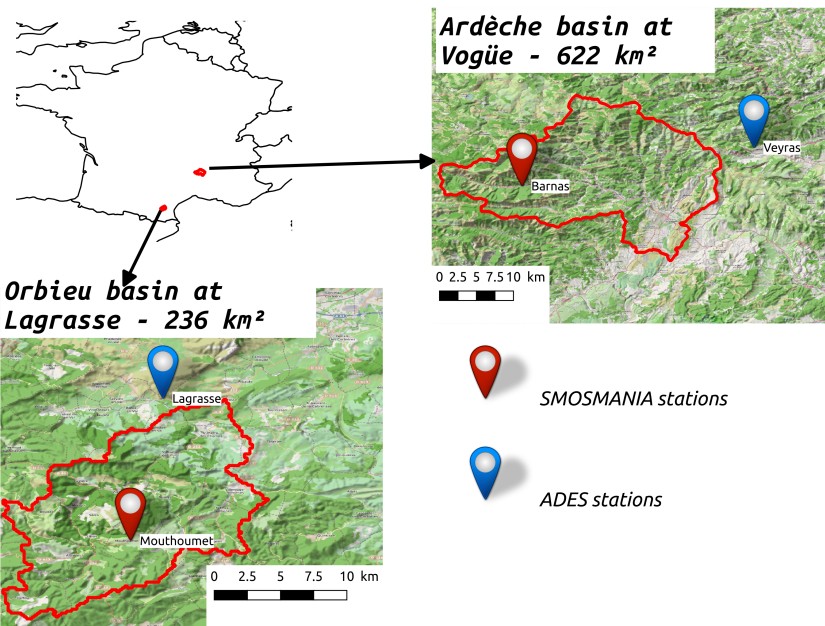

**Figure 1.** The two studied catchments located in the South of France: the Ardeche river at Vogue and the Orbieu river at Lagrasse. Monitoring networks: soil moisture (SMOSMANIA network stations) and the national groundwater ADES network stations (www.ades.eaufrance.fr).

### 2.2.2 The studied events

In this work, the ANTILOPE quantitatives precipitation estimates (QPE) (Champeaux et al., 2009) are used for precipitation
estimation. The ANTILOPE-QPE are based on a fusion between the radar data provided by the operational radar network
ARAMIS (Tabary, 2007) and the measurements at pluviometers, spatialised by krigging method. ANTILOPE-QPE precipitation are available on the hourly time step, at the kilometric resolution. The critized observed discharges at the outlet of the two catchments are taken from the hydrometric French database (www.hydro.eaufrance.fr). Table 1 presents the characteristics of the studied event.


Three flash flood events are considered for each catchments over the 2017-2019 period. The heterogeneity of the studied events has to be noted: for the Orbieu catchment, the extreme event of October 2018 represents the historical maximum for this region, with well known dramatic damages to infrastructures and populations. This flood has the particularity to be extremely fast, with about two hours between the precipitation peak and the discharge peak at the Lagrasse station. This response time
appears to be faster than the response time regularly considered for this station (about 5 hours). On the opposite, the two other events considered for the Orbieu catchment, in February and Mars 2017, represent relatively small floods, with return periods of five years and two years, respectively. For the Ardeche catchment, the 2018 autumn has the particularity to present a serie of intermediate flood events. For this period, the damages have mainly been induced by the duration of the flooding period.





For the event defined for this study (November 2018, 22nd to 28th), the precipitation amounts do not represent extreme value,
however, flood damages have been noticed during this period. In addition, different hydrological responses can by distinguished
for spring or autumn seasons, due to different soil and vegetation conditions, possible snow contribution and meteorological
antecedents. This variety in the structures of the six events considered for this study represents both a robustness guaranty and
a challenge for the modeling exercise.

**Table 1.** The six events considered in this work for the Ardeche at Vogue and the Orbieu at Lagrasse catchments, with cumulated volume
(Precip.) and maximal intensity ($I^{pr}_{max}$) of ANTILOPE-QPE precipitation, maximal hourly observed discharge ($Q^{obs}_{max}$). The stars indicate
the return period of the flood: (*) for a 2-years, (**) for a 5-years, and (***) for a 100-years return period. The given dates and duration are
the ones considered for the hydrological simulations. S.M. is the initial soil moisture provided by the SAFRAN-ISBA-MODCOU chain for
the first day of the simulations, on average over the catchment.

| | Ardeche catchment | | | Orbieu catchment | | |
|---|---|---|---|---|---|---|
| Event | Ev 03 2018* | Ev 11 2018** | Ev 04 2019* | Ev 02 2017** | Ev 03 2017* | Ev 10 2018*** |
| Dates | 09-20/03 | 22-28/11 | 23-29/04 | 10-18/02 | 23-28/03 | 14-19/10 |
| Duration | 11days | 6days | 6days | 8days | 6days | 4days |
| Precip. | 170 mm | 98 mm | 146 mm | 79 mm | 58 mm | 193 mm |
| $I^{pr}_{max}$ | $11\ mm.h^{-1}$ | $9\ mm.h^{-1}$ | $12\ mm.h^{-1}$ | $5\ mm.h^{-1}$ | $7\ mm.h^{-1}$ | $24\ mm.h^{-1}$ |
| $Q^{obs}_{max}$ | $580\ m^3.s^{-1}$ | $627\ m^3.s^{-1}$ | $513\ m^3.s^{-1}$ | $181\ m^3.s^{-1}$ | $99\ m^3.s^{-1}$ | $448\ m^3.s^{-1}$ |
| S.M. | 57.62 % | 62.69 % | 50.81 % | 55.5 % | 53.8 % | 47.83 % |

## 2.3 Soil moisture products available

### 2.3.1 The SAFRAN-ISBA-MODCOU products

The SAFRAN-ISBA-MODCOU operational modeling chain (SIM) (Habets et al., 2008) uses the ISBA surface scheme, cou-
pled with the MODCOU hydrological model for underground flows and forced by the SAFRAN atmospheric reanalysis. SIM
outputs are available since 1958, on an hourly basis, on a regular mesh at the 8-km resolution. In particular, SIM provides
moisture data for the root layer of the soil. This work uses the outputs of two available versions of SIM: 1) SIM1, which uses
the force-restore version of ISBA, ISBA-3L (Noilhan and Planton, 1989; Noilhan and Mahfouf, 1996); and 2) SIM2, which
uses the diffusive version of ISBA, ISBA-DIF (Decharme et al., 2011), with a vertical soil column discretization into a maxi-
mum of 14 layers. In ISBA-3L, the root zone moisture corresponds to the humidity of the second soil layer. In ISBA-DIF, the
humidity of the root zone is considered as the sum of the humidities of the ISBA-DIF layers between 10 cm and 30 cm deep
for this specific study. The daily soil humidities of SIM correspond to the value at 06 UTC each day. In this work, the root zone
moisture provided by the SIM1 product is used for the initialization of the soil saturation in MARINE, as it is the product used
by Douinot (2016) and Garambois (2012) to calibrate the MARINE model. The SIM2 soil moisture data is compared to the
MARINE soil moisture outputs.



### 2.3.2 The LDAS-Monde product

LDAS-Monde (Albergel et al., 2017) assimilates satellite derived data into the ISBA land surface model. It uses the ISBA-A-gs (Calvet et al., 1998) model, the $CO_2$-responsive version of ISBA. The diffusive version of ISBA (ISBA-DIF) is used. ISBA-A-gs allows to simulate photosynthesis and fluxes of $CO_2$. In addition, LDAS-Monde assimilates LAI (Leaf Index Area) data provided by the European service Copernicus Global Land (CGLS), with a sequential assimilation algorithm (Simplified Extended Kalman Filter). The contribution of the assimilation of satellite data for the simulation of surface fluxes has been tested for various application cases, in particular over Europe and France by Fairbairn et al. (2017), Leroux et al. (2018), Dewaele et al. (2017) and Barbu et al. (2011). In this work, the version of LDAS-Monde which uses the AROME atmospheric model outputs for the atmospheric forcing of the model is used (Albergel et al., 2018; Bonan et al., 2020). These AROME-forced outputs are available since July 2017, at the 2.5 kilometer resolution and at three-hour time steps.

### 2.3.3 Satellite derived products

Various products derived from remote imagery are available for soil moisture estimation, at various spatial and temporal scales. In particular, the relevance of five products is investigated for this study. Table 2 summarizes the investigated products and their main characteristics.

**Table 2.** Investigated satellite derived soil moisture products and their main characteristics: data produced, provided variable, spatial resolution, satellite imagery employed and associated average uncertainties when provided. NA stands for Not Applicable.

| Shortname | Producer | Variable | Spatial resol. | Satellite source | Uncertainty | Reference |
|---|---|---|---|---|---|---|
| CGLS SWI | CGLS | SWI | 1 km | Sentinel-1, MetOp/ASCAT | NA | (Bauer-Marschallinger et al., 2018a) |
| CGLS SSM | CGLS | SSM | 1 km | Sentinel-1 | 8% | (Bauer-Marschallinger et al., 2018a) |
| THEIA VHSR | THEIA-Land | SSM | 1 km | Sentinel-1, Sentinel-2 | NA | El Hajj et al. (2017) |
| SMOS-IC | INRA-CESBIO | SSM | 25 km | SMOS L3 | 5% | Fernandez-Moran et al. (2017) |
| ESA CCI | ESA | SSM | 25 km | AMI-WS, MetOp/ASCAT | 3% | Dorigo et al. (2015, 2017) |

- The Copernicus Global Land Service (CGLS) provides both Surface Soil Moisture (SSM) and Soil Water Index (SWI) values at the 1-km spatial resolution and at the daily time step (Bauer-Marschallinger et al., 2018a). The SWI product combines the Sentinel-1/C-SAR band data and the MetOp/ASCAT data, in accordance with the algorithm presented by Bauer-Marschallinger et al. (2018b), whereas the SSM product is derived from only the Sentinel-1/C-SAR band data. In this work, the SWI values provided for the top 5 cm soil are considered. The uncertainties for the CGLS SSM are computed by adding the different sources of uncertainty occurring in the product preparation and they represent about 8% of the SSM values. No uncertainties estimation is provided for the SWI product.





• The soil moisture with very high spatial resolution product (VHSR) (El Hajj et al., 2017), provided by the THEIA-Land pole (www.theia-land.fr), offers soil moisture maps with a 6-days frequency and at the sub-parcel scale on several sites in France, in Europe and around the Mediterranean basin. The THEIA-Land VHSR soil moisture product exploits the Sentinel-1 radar and Sentinel-2 optical Copernicus image series, following a neural networks signal inversion algorithm. The extent of the two studied basins is globally covered by this product. However, the footprints of the images being

variable depending on the dates, the whole catchments are not covered for all dates. The amount of gaps in this product is significant: only 12 images are available over the studied events. In particular, no data are available over the Ardeche catchment for the studied dates.

• The SMOS-IC product (Fernandez-Moran et al., 2017) provides daily SSM at the 25-km resolution. The SMOS-IC soil moisture are derived from the SMOS remote data, based on the algorithm presented by Wigneron et al. (2007). This

method uses the new calibrated values of the soil roughness and effective scattering albedo parameters presented by Li et al. (2020). The uncertainties associated with the SMOS-IC product are estimated through the TB-RMSE index, presented by Al-Yaari et al. (2019) and represent about $5\%$ of the SMOS-IC SSM values.

• The ESA CCI product provides surface soil moisture datasets at daily temporal time step and 25 km spatial resolution. In this product, the AMI-WS and MetOp/ASCAT/C-band data are merged with several radiometer soil moisture products,

along the algorithm presented by Wagner et al. (2012). The uncertainties associated with the ESA CCI SSM product is considered as the variance of the dataset, estimated through triple collocation analysis. Uncertainties represent about $3\%$ of the ESA CCI SSM values.

Figure 2 jointly displays the catchment average for these products over the studied events, as well as their respective fraction of missing values. The impact of the spatial resolution on the spatially averaged values can be clearly noticed. The coarse

resolution (e.g. 25 km and 30 km resolution) SMOS-IC and ESA CCI soil moisture products appear to be overall lower than the products at the kilometric resolution (CGLS and THEIA-Land VHSR). In addition, the ESA CCI product is known to provide globally wetter SSM than the SMOS-IC product, as mentioned by Dong et al. (2020). However, it is to be noted that this products inter-comparison is mainly informative regarding the products temporal dynamics but their respective biases cannot be directly compared, mainly for two reasons: i) the compared variables are not necessarily commensurable (i.e. SSM

and SWI); ii) the soil depth considered in each product for the SSM estimation might differ.

Important discrepancies are observed in the temporal dynamics for the different product. Since the study area is rather small, no validation of these products at the very local scale is available and the relatively low uncertainties estimates do not allow to explain these differences (see table 2). As no particular temporal behavior can be distinguished among the five product, the

choice has been done for this work to particularly focus on the product that offered the most important data availability and the finest spatial resolution. The amounts of missing values for the SMOS-IC and the THEIA-Land VHSR products, and also for the CGLS SSM products are too important for these data sources to be reliably used. On the contrary, the CGLS SWI product presents a good data availability, despite some events being less covered than others (e.g. March 2018 or November





2018 over the Orbieu catchment). In this product, the number of informative pixels per catchment for the studied cases is

greater than 14% of the catchment area. Consequently, in this work, the CGLS SWI product is taken into account to perform

the comparison with the soil moisture simulated in MARINE. Nevertheless, this literature exploration of the data available for

soil moisture description illustrates the difficulty to estimate surface soil moisture based on satellite data at small catchment

scale ($\sim 100 km^2$).

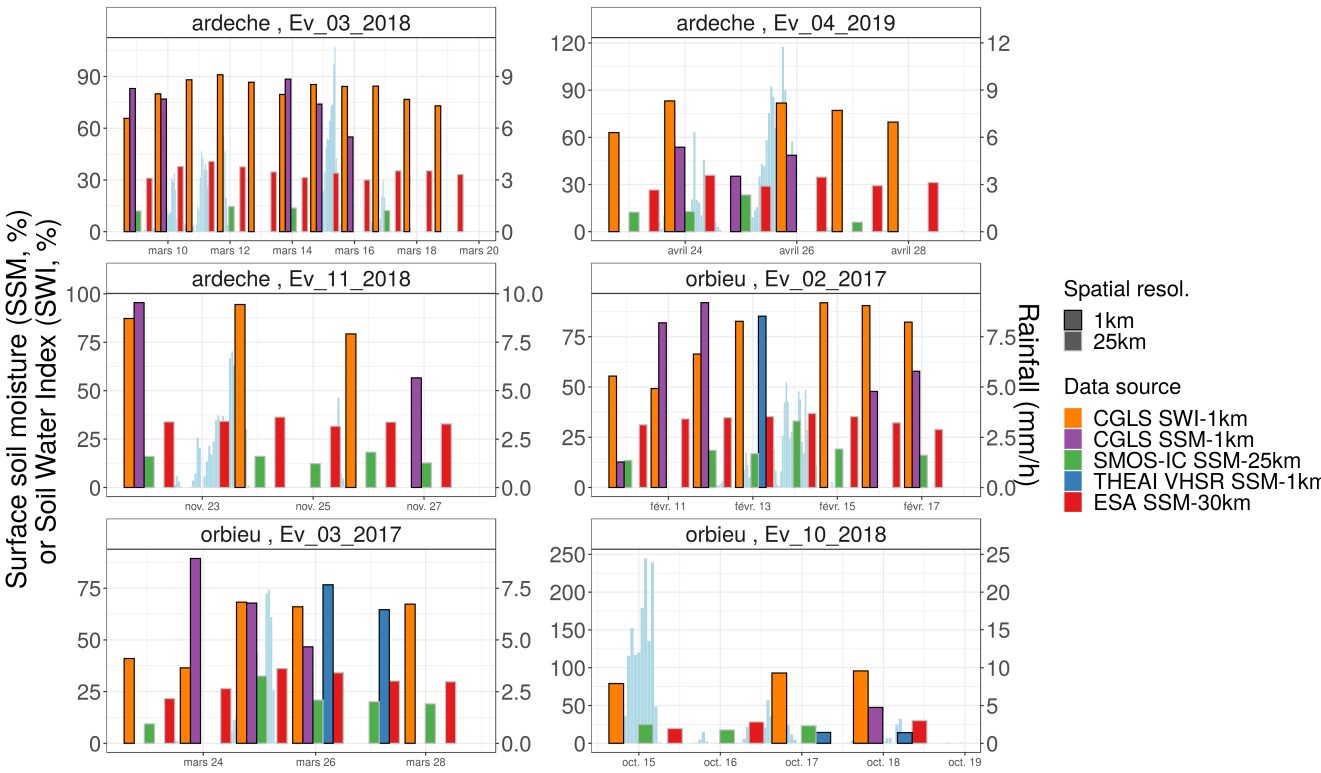

**Figure 2.** Daily values of Surface Soil Moisture (SSM) or Soil Water Index (SWI) provided by the CGLS, SMOS-IC, THEIA-Land VHSR and ESA CCI products (left axis), along with associated ANTILOPE precipitation (right axis),on average over the two studied catchments during the six simulated events.

### 2.3.4   The SMOSMANIA network

The SMOSMANIA project (Soil Moisture Observing System Meteorological Automatic Network Integrated Application, Cal-

vet et al. (2007); Parrens et al. (2012)) provides soil moisture measurements for 21 stations of the automatic ground station

network of Météo-France (the RADOME network), along a 400 km Mediterranean-Atlantic transect in southwestern France.

Each SMOSMANIA station is equipped with four ThetaProbes ML2X instruments forming a soil profile at the depths 5, 10,

20, 30 cm. Volumetric soil moisture is recorded at each depth and data are transmitted each 15 minutes since 2006 for all





the stations. Two stations are considered for this work: the Mouthoumet station, located inside the Orbieu at Lagrasse catchment, and the Barnas station, located inside the Ardeche at Vogue catchment. For these two stations, soil moisture profiles are available over the whole 2017-2019 period. The sensors calibrations are regularly checked and the vertical variability of soil properties is taken into account for these calibrations.

### 2.3.5 The ADES piezometric network

The ADES database (Access to Data on Groundwater, www.ades.eaufrance.fr), coordinated by the French National Geological Survey (BRGM), provides piezometric level measurements throughout France. One point of measurement is available for each of the two studied catchment. Figure 1 shows the location of the two measurement points. For the Orbieu catchment, the water table is 110 $km^2$ large and 1849 $km^2$ large for the Ardeche catchment. The measurements are available at the daily time step and the daily value represents the maximum of the water level measurements in 24 hours. In this work, the relative underground

water level with respect to the measurement mark is compared to the water content of the deep layer simulated with SSF-DWF model.

## 3 Methods

### 3.1 Comparison protocol

#### 3.1.1 Choice of layers for the LDAS-Monde soil moisture

Figure 3 presents the spatial average of the soil moisture, for each catchment and for each of the eleven soil layers described in the LDAS-Monde product. Two behaviors can be distinguished for the different layers: for the five superficial layers, a fast-responding soil moisture and a more stable soil moisture, with a slower response to precipitation and narrower amplitude range for the deeper layers. Moreover, the diurnal cycle of solar radiation significantly influences up to the fifth layer, i.e. up to 40 cm deep. In addition, over the two studied catchments, the spatial patterns of soil moisture are similar for the eleven

layers. Indeed, the spatial distribution of soil moisture is mainly controlled by the soil texture, which is considered as vertically uniform in the ISBA-A-gs model. Consequently, the choice is made in this work to synthesize the eleven LDAS-Monde layers as three average layers: the surface layer (average of layers 1 to 5), the deep layer (average of layers 6 to 11), and the total layer (average of all the 11 layers). Thus, the surface layer represents depths from 0 cm to 40 cm and the deep layer represents depths from 40 cm to 300 cm. Concerning the comparison between the MARINE simulation and LDAS-Monde, for the base and SSF

models, which use a one layer soil discretization, the MARINE soil moisture is compared to the moisture of the surface layer, noted $HU_{surf}$. For the SSF-DWF model, which uses a two-layers soil discretization, the moisture of the MARINE upper layer is compared to LDAS-Monde surface layer, and the moisture of the MARINE deep layer is compared to the LDAS-Monde deep layer (noted $HU_{deep}$). The total average LDAS-Monde layer is used for overall comparison.



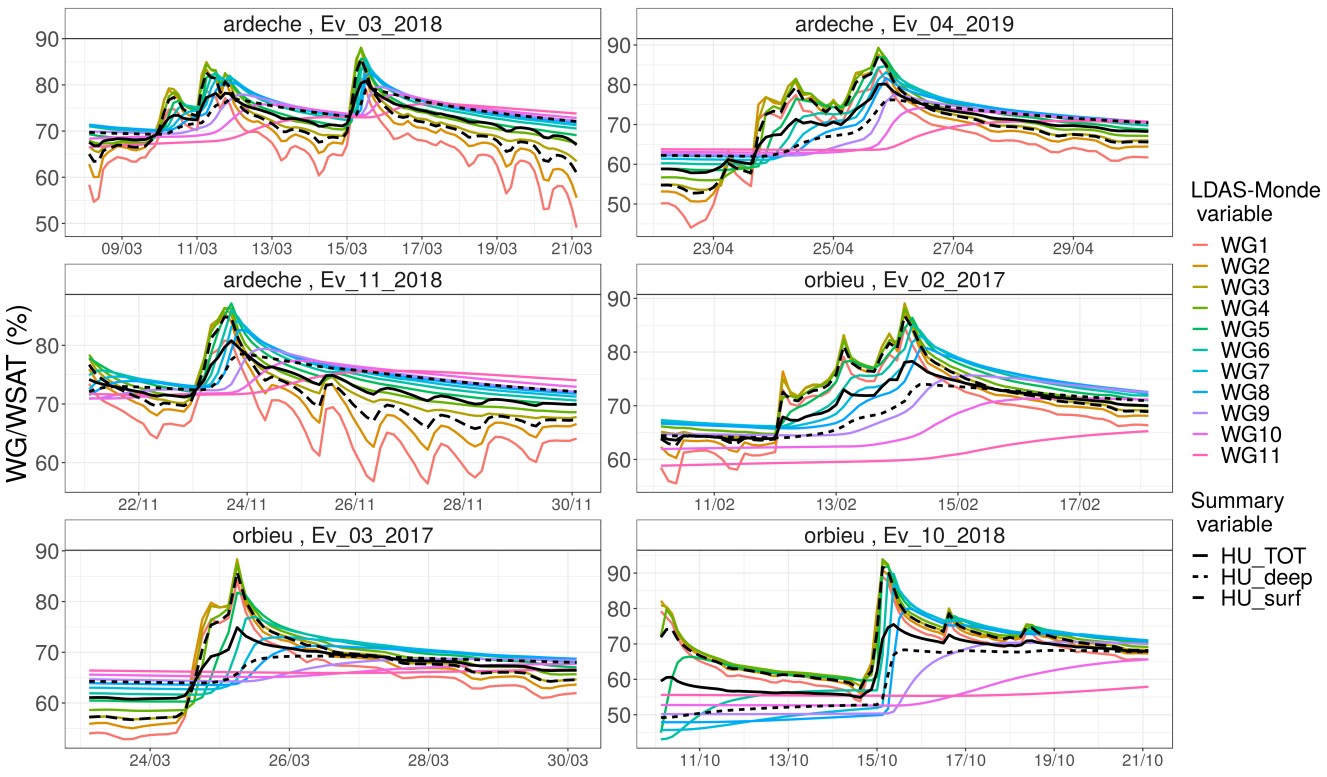

**Figure 3.** Soil moisture (%) for the 11 soil layers described in LDAS-Monde and summary variables $HU_{surf}$ (average of the layers 1 to 5), $HU_{deep}$ (average of the layers 6 to 11) and $HU_{tot}$ (average of the layers 1 to 11), in average per catchment for the six studied events.

### 3.1.2 Method for comparing gridded data to SMOSMANIA observations

The SMOSMANIA observation network provides valuable information for the upper soil water content. However, it raises the issue to compare point measurements to the gridded simulated soil moisture. Various strategies might be used to face this issue, among which averaging at a large time scale (Tramblay et al., 2010; Fuamba et al., 2019). In this study, considering the fast-evolving processes involved, we choose to maintain the hourly time step for soil moisture analysis. The important spatial variability of the soil moisture is then taken into account by spatial averaging the gridded simulated values around the

measurement point. In order to consider equivalent surfaces for the grids simulated in MARINE and provided by the LDAS-Monde and CGLS data, the MARINE soil moisture maps are averaged on a 1 $km^2$ area around the measurement point. In addition, the MARINE drainage network is excluded from this average area, because the physic of the soil saturation in the drainage network is not commensurable with its physics over hillslope meshes. This leads to exclude 4 meshes over 16 from the average area for the Ardeche catchment, and no mesh for the Orbieu catchment.





### 3.1.3 Indices


The performance of the simulated discharges are estimated at the hourly time step through the usual Nash Sutcliffe Efficiency criteria (NSE) and also through the LNP index, defined by Roux et al. (2011) as in equation 1, where $Q^{obs}$ ($Q^{obs}_{max}$) and $Q^{sim}$ ($Q^{sim}_{max}$) represent the (maximal) observed and simulated discharged, respectively, and $T_{concentration}$, the concentration time of the catchment. The advantage of the LNP index is to give equal weight to the NSE values (first term), to the peak value estimation (second term) and to the timing of the peak simulation (third term). LNP appear to be a integrative criteria well-suited for flash flood modelling (Lovat et al., 2019).


$$LNP = \frac{1}{3}.(1 - \frac{\sum_{i}(Q^{sim}_i - Q^{obs}_i)^2}{\sum_{i}(Q^{obs}_i - \overline{Q^{obs}_i})^2}) + \frac{1}{3}.(1 - \frac{|Q^{sim}_{max} - Q^{obs}_{max}|}{Q^{obs}_{max}}) + \frac{1}{3}.(1 - \frac{|T^{sim}_{max} - T^{obs}_{max}|}{T^{obs}_{concentration}}) \tag{1}$$

The comparison of the soil moisture simulated in MARINE and provided by LDAS-Monde is performed at the catchment scale using the relative bias and the Kendall correlation over values averaged at the catchment scale. In addition, the spatial dynamics of the simulated soil moisture are quantified using the spatial moments $\delta_1$ and $\delta_2$ defined by Zoccatelli et al. (2011). The $\delta_1$ and $\delta_2$ moments take into account the distance of each grid cell to the drainage network and they allow to represent both the overall location of the soil moisture field with respect to the outlet and the number of modes (i.e concentration points in this case) of the field. The closer of 1 are the $\delta_1$ values, the more centred around the centroid of the catchment is the field. Values of $\delta_1$ lower that 1 mean that the field get closer from the outlet, whereas values higher that 1 characterize a field overally located on the highest areas of the catchment. The closer of 1 are the $\delta_2$ values, the more uniform is the distribution of the field. Values of $\delta_2$ lower that 1 represent an unimodal distribution and values of $\delta_2$ higher that 1 mode likely represent a multimodal distribution. Despite being initially defined by Zoccatelli et al. (2011) to characterize rainfall fields, the $\delta_1$ and $\delta_2$ moments also appear to be particularly relevant when applied to soil moisture fields.



### 3.2 Model set up


#### 3.2.1 Parametrization and precipitation forcing

The MARINE model requires the definition of i) the digital elevation model (DEM), ii) soil survey data to compute the hydraulic and storage properties of the soil and iii) land-use data to configure the surface roughness parameters. The IGN-25 m DEM is used in this work. The soil depths and soil texture maps are taken from the INRA soil data base for the Ardèche and Languedoc-Roussillon regions (Robbez-Masson et al., 2000). The parameters of the pedotransfer function are computed based on the USDA soil classification (Spaargaren, 1995). Land cover is provided by the Corine Land Cover 2006 data base (Aune-Lundberg and Strand, 2010). The model is set up over a regular mesh, with a 200 m spatial resolution for the Orbieu catchment and a 250 m resolution for the Ardeche catchment. The model computation time step is 5 minutes and results are aggregated at the hourly time step. This study uses the calibration of MARINE provided by Garambois et al. (2015) for the Orbieu catchment






and by Douinot (2016) for the Ardeche catchment. The ANTILOPE QPE data are used as hourly precipitation input for the

MARINE model, available at the kilometric resolution. Figure 4 presents the IGN-25 m DEM and the soil depth maps used for

the two studied catchments. Table 3 presents the calibrated parameter values obtained for each catchment by Douinot (2016)

and Garambois et al. (2015) and used in this work.

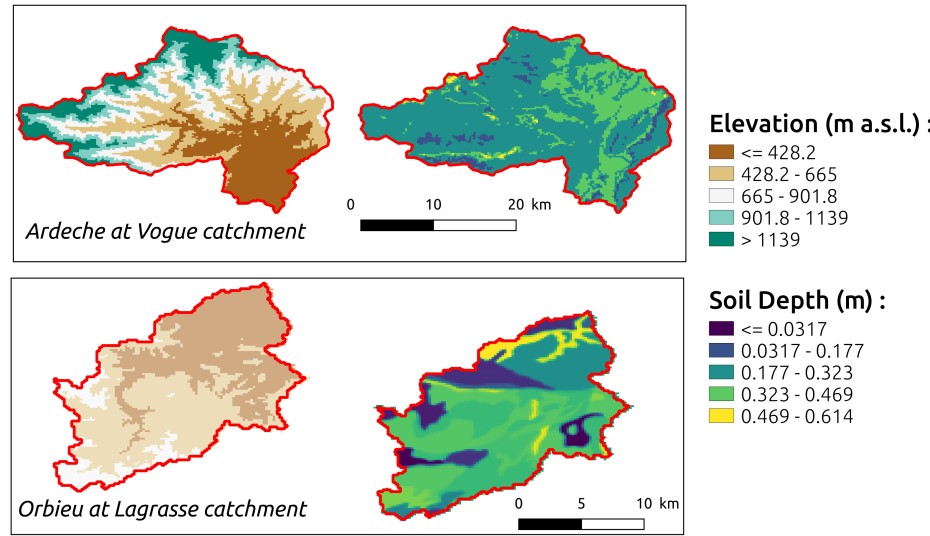

**Figure 4.** The IGN-25 m DEM and soil depth maps from the INRA soil data base used for MARINE parametrization for the two studied catchments.

**Table 3.** Calibrations obtained by Douinot (2016) and Garambois et al. (2015) for the Orbieu at Lagrasse and Ardeche at Vogue catchments: the multiplier coefficient for soil depth maps ($C_z$), the multiplier coefficient for the spatialized saturation hydraulic conductivity used in lateral flow modelling ($C_{kss}$) the multiplier coefficient for the spatialized hydraulic conductivity at saturation that is used in infiltration modelling ($C_{kga}$), two friction coefficients for low and high-water channels ($C_{D1}$ and $C_{D2}$), and deep layer depth for the SSF-DWF model ($C_z^{deep}$).

| Basin: | | Ardeche | Orbieu |
|---|---|---|---|
| Calibration: | | Douinot (2016) | Garambois et al. (2015) |
| $C_z$ | $(-)$ | 2.86 | 1.3 |
| $C_{kga}$ | $(-)$ | 1.34 | 15 |
| $C_{kss}$ | $(-)$ | 3241 | 10000 |
| $C_{D1}$ | $(m^{1/3}.s^{-1})$ | 14.4 | 9.1 |
| $C_{D2}$ | $(m^{1/3}.s^{-1})$ | 18.5 | 2 |
| $C_z^{deep}$ | $m$ | 1.42 | 0.51 |





### 3.2.2 Discharge simulation

Figure 5 presents the discharges at the outlets, simulated with MARINE using the base, the SSF or the SSF-DWF models together with the observed discharges during the flood events. Table 4 presents the associated LNP and Nash Sutcliffe Efficiency (NSE) performance criterias of the simulated discharges, referring to hourly observed discharges. The main effect of computing the transfers through the subsurface as a function of the volumetric soil water content instead of the water height (SSF model) is to flatten the overestimation of the simulated discharge during the flow rise, at the beginning of the events. This behavior

will be explained in the result section: there is no gradient of initial soil water content over the 8x8km SIM mesh and therefore smaller subsurface contribution at the beginning of the events in the SSF and SSF-DWF. However, in the SSF-DWF model, this dynamics is influenced by the contribution of the deep layer, itself mainly controlled by the parametrization of the thickness of this deep layer. Nevertheless, the calibrations of the three models clearly require to be improved in order to better simulate the discharges at the outlets, in particular for the Orbieu catchment and for the SSF-DWF model. However, since this paper focuses

on comparing the soil moisture dynamic simulation according to the soil physics considered in the model, and considering that the variety in the structures of the considered events (see section 1) is a limit to model performances, the calibration proposed by Douinot (2016) and Garambois et al. (2015) are directly applied to this work.

**Table 4.** LNP and Nash Sutcliffe Efficiency (NSE) performance criterias for discharges simulation at the outlet for the six studied events over the two catchments, for the base, the SSF and the SSF-DWF models, referring to hourly observed discharges.

| Ardeche catchment | | | | Orbieu catchment | | | |
|---|---|---|---|---|---|---|---|
| Event | Model | LNP | NSE | Event | Model | LNP | NSE |
| Ev 03 2018 | BM | 0.79 | 0.57 | Ev 02 2017 | BM | -0.36 | -2.46 |
| Ev 03 2018 | SSF | 0.63 | 0.24 | Ev 02 2017 | SSF | 0.26 | 0.38 |
| Ev 03 2018 | SSF-DWF | 0.49 | 0.09 | Ev 02 2017 | SSF-DWF | -0.09 | -1.28 |
| Ev 04 2019 | BM | 0.58 | -0.12 | Ev 03 2017 | BM | -3.55 | -11.27 |
| Ev 04 2019 | SSF | 0.26 | 0.75 | Ev 03 2017 | SSF | 0.25 | 0.23 |
| Ev 04 2019 | SSF-DWF | 0.15 | 0.69 | Ev 03 2017 | SSF-DWF | -1.62 | -5.93 |
| Ev 11 2018 | BM | 0.76 | 0.44 | Ev 10 2018 | BM | -0.43 | -2.28 |
| Ev 11 2018 | SSF | 0.57 | 0.15 | Ev 10 2018 | SSF | 0.26 | -0.31 |
| Ev 11 2018 | SSF-DWF | 0.73 | -0.37 | Ev 10 2018 | SSF-DWF | -0.19 | -1.56 |

## 4 Results and discussions

### 4.1 Comparison at the point measurement scale

Figure 6 puts together i) the soil moisture measurement at the four sensor depths for the Barnas (for the Ardeche catchment) and the Mouthoumet (for the Orbieu catchment) SMOSMANIA stations; ii) the soil moisture simulated with MARINE, on



**Figure 5.** Discharges at the outlets, simulated with MARINE using the base, the SSF and the SSF-DWF models, and observed discharges.

average over a 1-$km^2$ area over the station location (see section 3.1.2). For the simulations using the SSF-DWF soil model, the moisture of the surface layer is considered here; iii) the LDAS-Monde surface soil moisture $HU_{surf}$ for the 2.5 $km$x2.5 $km$ grid cell that contains the SMOSMANIA station; iv) the CGLS SWI when available for the 1 $km$x1 $km$ grid cell that

contains the SMOSMANIA station for the Orbieu catchment. No data are available for the grid cell that contains the station for the Ardeche catchment. Table 5 provides the Kendall correlations associated with the hourly time series presented on figure 6. The values in bold are the best correlation values between the SMOSMANIA measurements and the MARINE outputs or the LDAS-MONDE $HU_{surf}$ for each event.

Soil moisture simulated with the base model significantly differs from the simulations using the SSF and the SSF-DWF models: the soil layer empties faster with the base model, leading to a simulated soil moisture significantly lower with the base model than with the two other models. Overally for the simulated events, the simulated soil moisture and the SMOSMANIA measurements appears to be better correlated when using the SSF-DWF model rater than the base model or the SSF model.





The soil physics used in the SSF-DWF, i.e. the use of the volumic soil water content rate and the vertical discretization into two
layers, allows to enhance the soil moisture simulation for the surface layer, with respect to in-situ measurements. This point
will be developed by considering the catchment average of simulated soil moisture in the next section.

In addition, the soil moisture simulated for the surface layer with the SSF-DWF is globally higher than for the two other
models. This behavior can be explained by the fact that, for the SSF-DWF model, soil depths are taken from the INRA soil data
base, whereas for the base model and SSF model, a multiplicative, calibrated coefficient superior to 1 is applied. Consequently,
the depths considered for the surface layer are thinner in the SSF-DWF than in the base model and SSF model. The saturation
of the surface layer is then reached faster.

Besides, the LDAS-Monde $HU_{surf}$ appears to be globally satisfyingly correlated with the SMOSMANIA measurements,
with slightly different correlations for the four sensor depths. This shows that the dynamic of the LDAS-Monde $HU_{surf}$
variable is locally significant with in-situ surface soil moisture measurement. The reliability of the LDAS-Monde $HU_{surf}$
dynamic for surface soil moisture description can thus be considered as satisfying. On the contrary, the correlation between the
daily CGLS SWI values and both the MARINE outputs and the SMOSMANIA measurements appear to be low. However, a
more extensive study of the validity of this product at the local scale would be needed to draw further conclusions.

## 4.2 Comparison at the catchment scale

### 4.2.1 Water content of the surface layer

Figure 7 presents the soil moisture time series, on average per catchment, simulated with MARINE using the base, the SSF
or the SSF-DWF models, together with the catchment average of the LDAS-Monde $HU_{surf}$, the daily CGLS SWI values and
the daily SIM2 HU values (see section 2.3.1). When the SSF-DWF model is applied, the surface layer is considered here.
Table 6 presents the Kendall correlations associated with the hourly times series. The same observations as for the comparison
at the local scale can be drawn: both the dynamics and the amplitudes of the soil moisture simulated with the base model
significantly differ from the outputs of the two other models. When no precipitation happens, the soil drainage in the base
model is faster than for the SSF and the SSF-DWF models. In addition, the soil moisture simulated with the SSF-DWF model
is globally higher than the one simulated with the SSF model, on average per catchment. The soil moisture simulated with
the SSF-DWF model appears to be better correlated with the LDAS-Monde $HU_{surf}$ time series, for four of the six studied
events. Considering that the dynamics of the LDAS-Monde $HU_{surf}$ is of satisfying accuracy (see section 4.1), the SSF-DWF
model appear to improve the simulation of the dynamics of the surface layer moisture, compared to both the SSF and the
base models. This results appears to be particularly reliable, since it is observed both a the point measurement scale and at
the catchment scale. It can be physically explained by the fact that, in the SSF and the SFF-DWF models, the lateral transfers
are computed as a function of the volumic soil water gradients, whereas in the base model, they are computed as a function
of the water height gradient. Indeed, since the water height gradient between two cells depends on the slope between the cells



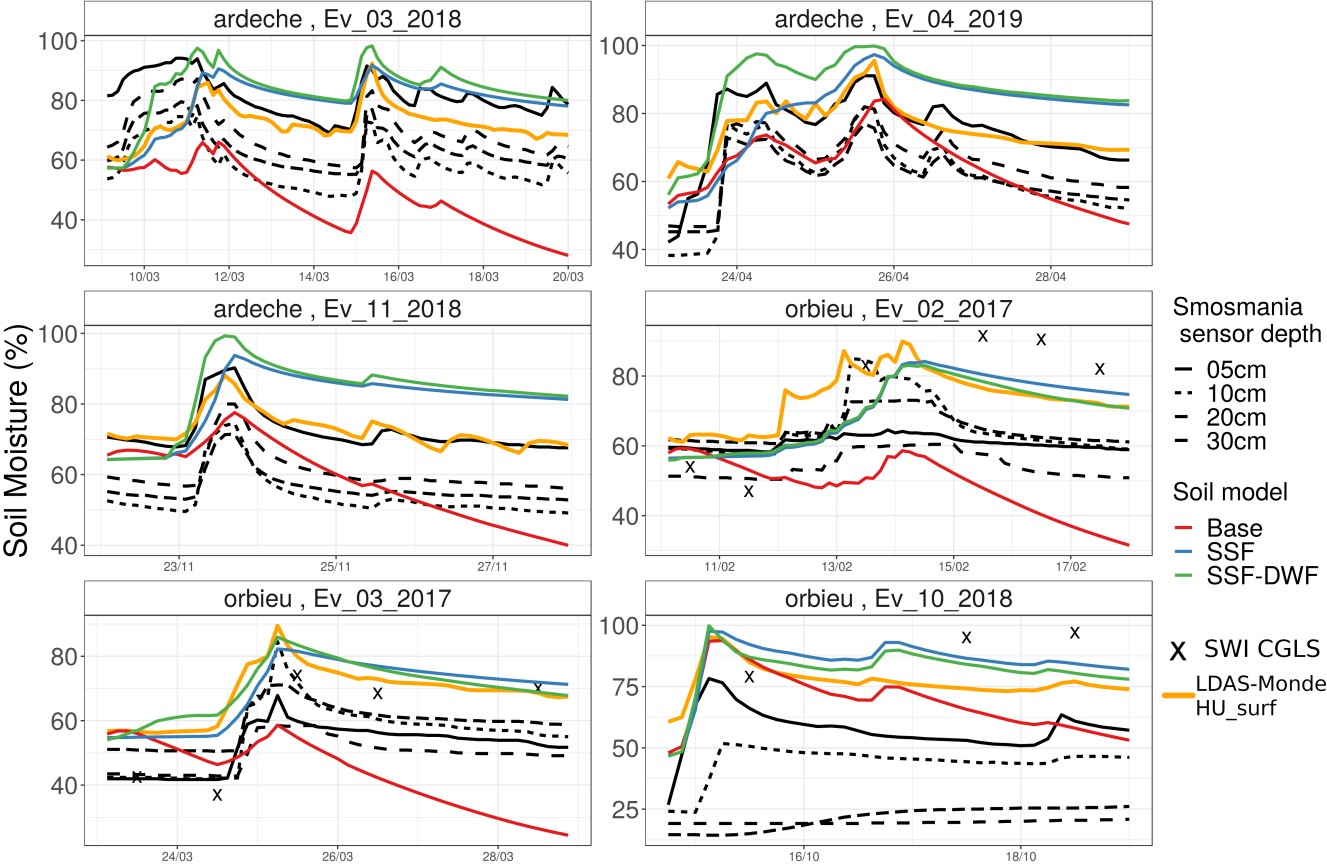

**Figure 6.** SMOSMANIA soil moisture measurement at the four sensor depths for the Barnas (Ardeche catchment) and the Mouthoumet (Orbieu catchment) stations, together with the soil moisture simulated with MARINE, the LDAS-Monde $HU_{surf}$ and the CGLS SWI when available at the measurement point location. For the MARINE simulations using the SSF-DWF soil model, the moisture of the surface layer is considered here.

and the cells textures, water height gradients are larger than volumic soil water gradient when no precipitation happens. Consequently, lateral flows based on the water height gradients are larger than lateral flows based on the volumic soil water gradient.

On overall, the temporal dynamics of the CGLS SWI, in average per catchment is more consistent with the SSF and SSF-DWF models outputs than with the base model output. In particular, for the events of February and March 2017 on the Orbieu catchment, the sharp decreases of the soil moisture simulated in the base model is not observed in the CGLS SWI values. In addition, for the event of Novembre 2018 on the Ardeche catchment, which is the longest of the studied events, the dynamic of the CGLS SWI is very consistent with the soil moisture simulated with the SSF and SSF-DWF models. Likewise, catchment

averages of the SIM2 HU values are also better correlated with the SSF and SSF-DWF models outputs than with the base





**Table 5.** Kendall correlations between Smosmania measurements at each depth and MARINE soil moisture simulated with each soil model or the LDAS-Monde $HU_{surf}$ . The values in bold are the best correlations between the SMOSMANIA measurements and the MARINE outputs or the LDAS-MONDE $HU_{surf}$ for each events.

| | | Orbieu catchment | | | Ardeche catchment | | |
|---|---|---|---|---|---|---|---|
| Soil model | Depth | Ev 02 2017 | Ev 03 2017 | Ev 10 2018 | Ev 11 2018 | Ev 03 2018 | Ev 04 2019 |
| Base | 05cm | 0.254 | 0.239 | **0.512** | 0.569 | 0.452 | 0.69 |
| Base | 10cm | 0.193 | 0.24 | 0.499 | 0.617 | 0.41 | 0.695 |
| Base | 20cm | 0.248 | 0.261 | -0.65 | 0.617 | 0.457 | 0.693 |
| Base | 30cm | 0.207 | 0.211 | -0.625 | 0.631 | **0.493** | 0.694 |
| SSF | 05cm | 0.457 | 0.76 | 0.354 | 0.476 | 0.122 | 0.368 |
| SSF | 10cm | 0.486 | 0.777 | 0.44 | 0.507 | 0.161 | 0.40 |
| SSF | 20cm | 0.518 | 0.736 | -0.435 | 0.571 | 0.19 | 0.416 |
| SSF | 30cm | **0.569** | 0.744 | -0.391 | 0.573 | 0.208 | 0.447 |
| SSF-DWF | 05cm | 0.488 | 0.83 | 0.303 | 0.622 | 0.379 | 0.808 |
| SSF-DWF | 10cm | 0.518 | **0.839** | 0.331 | 0.646 | 0.404 | 0.843 |
| SSF-DWF | 20cm | 0.544 | 0.808 | -0.4 | **0.698** | 0.427 | **0.855** |
| SSF-DWF | 30cm | 0.59 | 0.801 | -0.342 | 0.665 | 0.436 | 0.846 |
| $HU_{surf}$ | 05cm | 0.826 | **0.909** | **0.748** | 0.67 | 0.25 | 0.766 |
| $HU_{surf}$ | 10cm | **0.846** | 0.869 | 0.641 | **0.672** | 0.27 | **0.815** |
| $HU_{surf}$ | 20cm | 0.841 | 0.88 | -0.537 | 0.649 | 0.285 | 0.814 |
| $HU_{surf}$ | 30cm | 0.779 | 0.819 | -0.467 | 0.639 | **0.305** | 0.806 |

model output, despite the ranges of variation of the daily SIM2 HU values are narrower than the range for the CGLS SWI values.

Figure 8 presents maps of soil moisture simulated with the base, the SSF and the SSF-DWF models, and the maps of
LDAS-Monde $HU_{surf}$, for the example of the event of November, 2018 on the Ardeche catchment. The daily products are not presented here because the daily time step does not allow to represent the fast-evolving flood processes. Four time steps of the simulation are considered: first time step of the run, one time step during the flow rise, the peak flow hour and one time step in the flow decreasing. This example illustrates the results previously described: the saturation of the surface layer is faster reached for the SSF-DWF model than in the others. In addition, the spatial pattern of the soil moisture simulated with MA-
RINE appears to consistent with LDAS-Monde $HU_{surf}$ maps. An other interesting result is that the soil moisture initialization pattern seems to be vanished after a few rainy simulation time step. These results are also observed for the other events, not presented here.





Figures 9 and 10 present the $\delta_1$ and $\delta_2$ spatial moments computed for the MARINE soil moisture outputs, for the LDAS-Monde $HU_{surf}$ and for the CGLS SWI at the daily time step. Since no lateral transfers are represented in the LDAS-Monde and the CGLS SWI product, the MARINE drainage network is used to compute the spatial moments for both of them. The distinction between the base model outputs and the SSF and SSF-DWF model outputs can still be made. The general behavior of the $\delta_1$ spatial moment when computed on the soil moisture is that the $\delta_1$ increases when precipitation happens and then decreases at a variable rate. Indeed, precipitation that waters the catchment are doomed to flow toward the outlet. The $\delta_1$ time series obtained with the base model appear to be significantly lower than for the SSF and the SSF-DWF models. This can be explained by the faster emptying of the upper soil layer in the base model than in the other two models. Indeed, faster lateral transfers from each cell to its downhill cell lead to soil moisture distribution overally higher around the outlet at each time step.

The general behavior of the $\delta_2$ spatial moment is that the $\delta_2$ decreases with precipitation, with soil moisture fields more centered around the area of maximum rainfall, and then increases with the spread of the soil moisture fields along the drainage network. The $\delta_2$ values for the SSF and SSF-DWF models are globally closer to 1 than for the base model. Indeed, since the soil saturation is globally higher for the SSF and SSF-DWF models (see figure 7), the difference between the soil saturation and saturation in the drainage network (i.e. 100%) is stronger for the base model than for the other two models. This leads to soil moisture fields more uniform for the SSF and SSF-DWF models than for the base model. This result is particularly observed for the Orbieu catchment.

Both the $\delta_1$ and $\delta_2$ spatial moments computed for the LDAS-Monde $HU_{surf}$ are globally closer to 1 than when computed for the MARINE outputs. Indeed since the spatial resolution is the LDAS-Monde $HU_{surf}$ is 2.5x2.5 $km^2$, whereas it is 200x200 m or 250x250 m for the MARINE simulations, the spatial variability of the LDAS-Monde $HU_{surf}$ is lower than for the MARINE outputs. The $\delta_1$ and $\delta_2$ spatial moments computed for the CGLS SWI are very close to 1, with tiny variations. This can be explained not only by the spatial resolution coarser than for the MARINE outputs but also by the important amount of missing pixel in this data source, in particular for the Ardeche catchment. The computation of spatial moments for the CGLS SWI might not lead to robust conclusions.

**Table 6.** Kendall correlations between LDAS-Monde and MARINE soil moisture, on average per catchment, for each soil model.

| Soil model | LDAS-Monde | Orbieu catchment | | | Ardeche catchment | | |
|---|---|---|---|---|---|---|---|
| | | Ev 02 2017 | Ev 03 2017 | Ev 10 2018 | Ev 11 2018 | Ev 03 2018 | Ev 04 2019 |
| Base | $HU_{surf}$ | 0.092 | 0.19 | **0.647** | **0.642** | 0.534 | 0.623 |
| SSF | $HU_{surf}$ | 0.581 | 0.752 | 0.601 | 0.402 | 0.332 | 0.406 |
| SSF-DWF | $HU_{surf}$ | **0.6** | **0.867** | 0.59 | 0.512 | **0.647** | **0.724** |





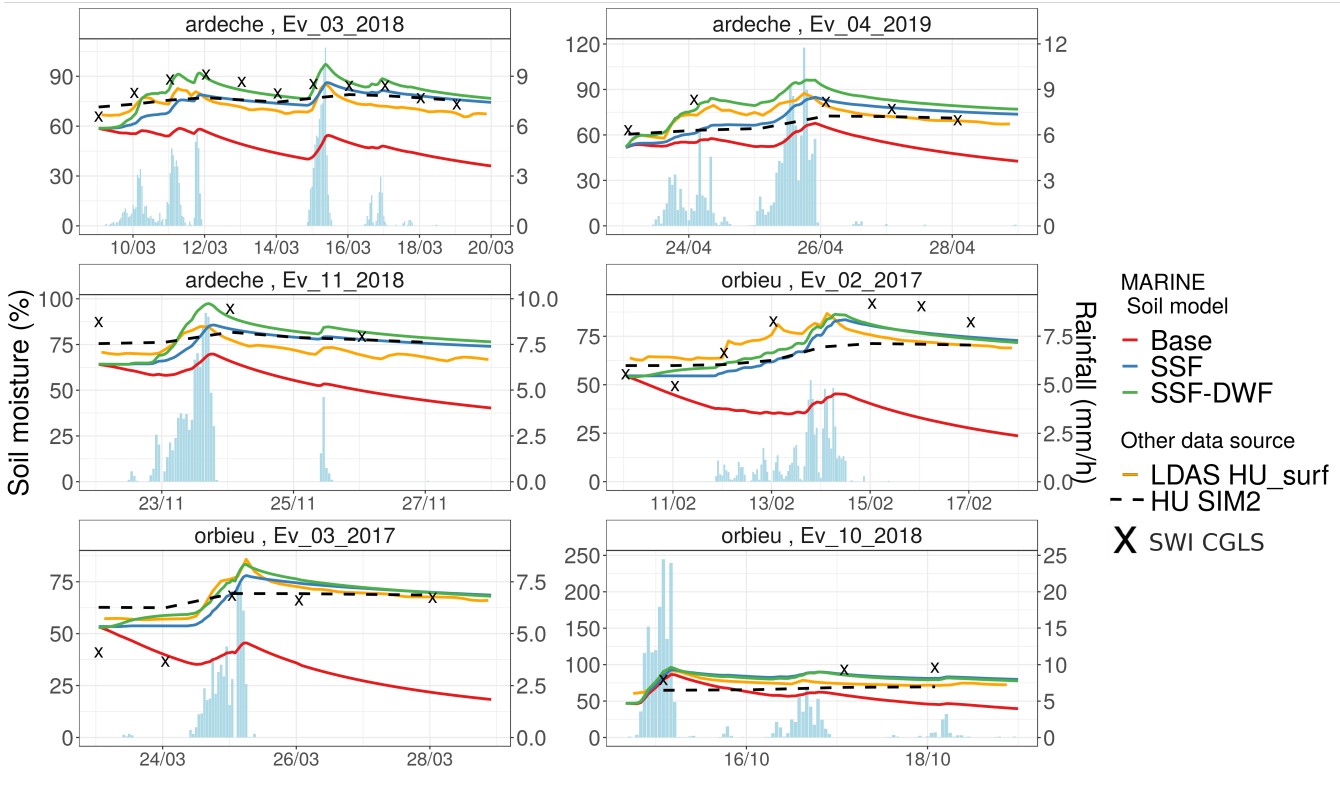

**Figure 7.** Soil moisture time series, on average per catchment, simulated with MARINE using the base, the SSF or the SSF-DWF models, and LDAS-Monde $HU_{surf}$ and SWI CGLS values, in average per catchment.

### 4.2.2 Water content of the deep layer

Figure 11 presents the soil moisture simulated for the deep layer with the SSF-DWF model, together with the LDAS-Monde $HU_{deep}$ time series, on average per catchment. The piezometric levels recorded at the measurement point of the ADES network for each catchment are also represented on this figure. Table 7 presents the Kendall correlations between the SSF-DWF deep layer moisture and the LDAS-Monde $HU_{deep}$.

**Table 7.** Kendall correlations between LDAS-Monde and MARINE deep layer moisture for the SSF-DW model.

|  |  | Orbieu catchment | | | Ardeche catchment | | |
|---|---|---|---|---|---|---|---|
| Soil model | LDAS-Monde | Ev 02 2017 | Ev 03 2017 | Ev 10 2018 | Ev 11 2018 | Ev 03 2018 | Ev 04 2019 |
| SSF-DWF | $HU_{deep}$ | -0.401 | -0.258 | -0.005 | 0.757 | 0.642 | 0.869 |





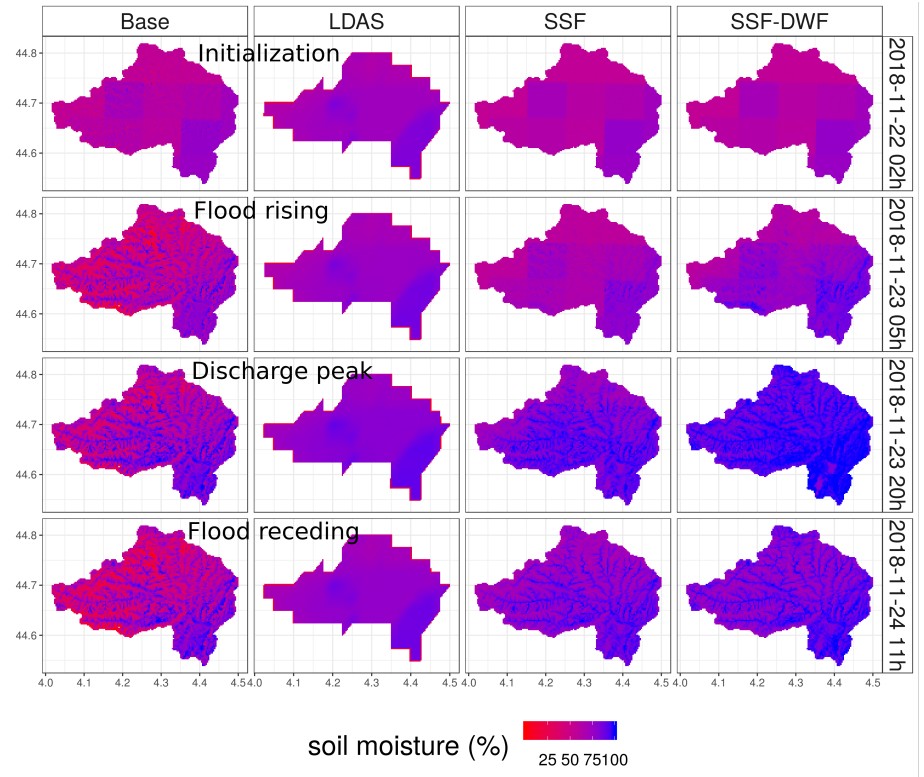

**Figure 8.** Maps of simulated soil moisture, for the example of the event of November, 2018 on the Ardeche catchment. MARINE simulation output with the base, the SSF and the SSF-DWF models are presented, and also the LDAS-Monde $HU_{surf}$ maps. Four time steps of the simulation are considered: first time step of the run, one time step during the flow rise, the peak flow hour and one time step in the flow decreasing.

For the Ardeche catchment, the simulated deep layer moisture is well correlated with the LDAS-Monde $HU_{deep}$, with Kendall correlations between 6.4 and 8.7. This result enhance the reliability of the deep layer calibration in the SSF-DWF model for the Ardeche catchment. However, for this catchment, as the extend of the water table (1849 $km^2$) is large compared to the area impacted by extreme precipitation, the response of the piezometric level of the water table to the precipitation event is small. Then, these measurements can not be used to assess the simulated moisture of the deep layer at the catchment scale.


    Furthermore, for the events over the Orbieu catchment, the simulated deep layer moisture appears not to be consistent with the LDAS-Monde $HU_{deep}$, in particular for the two events of February and March 2017. For the strong event of October 2018 on the Orbieu catchment, the sharp increasing of the deep soil moisture at the end of the rainfall event is observed in both the SSF-DWF model and in the LDAS-Monde $HU_{deep}$. The calibration of the deep layer in the SSF-DWF model for the Orbieu

catchment leads to an emptying of deep soil faster than for the LDAS-Monde $HU_{deep}$ variable. The simulation of the deep layer water content strongly depends on the calibration of the deep layer thickness, the deep layer porosity and the vertical and

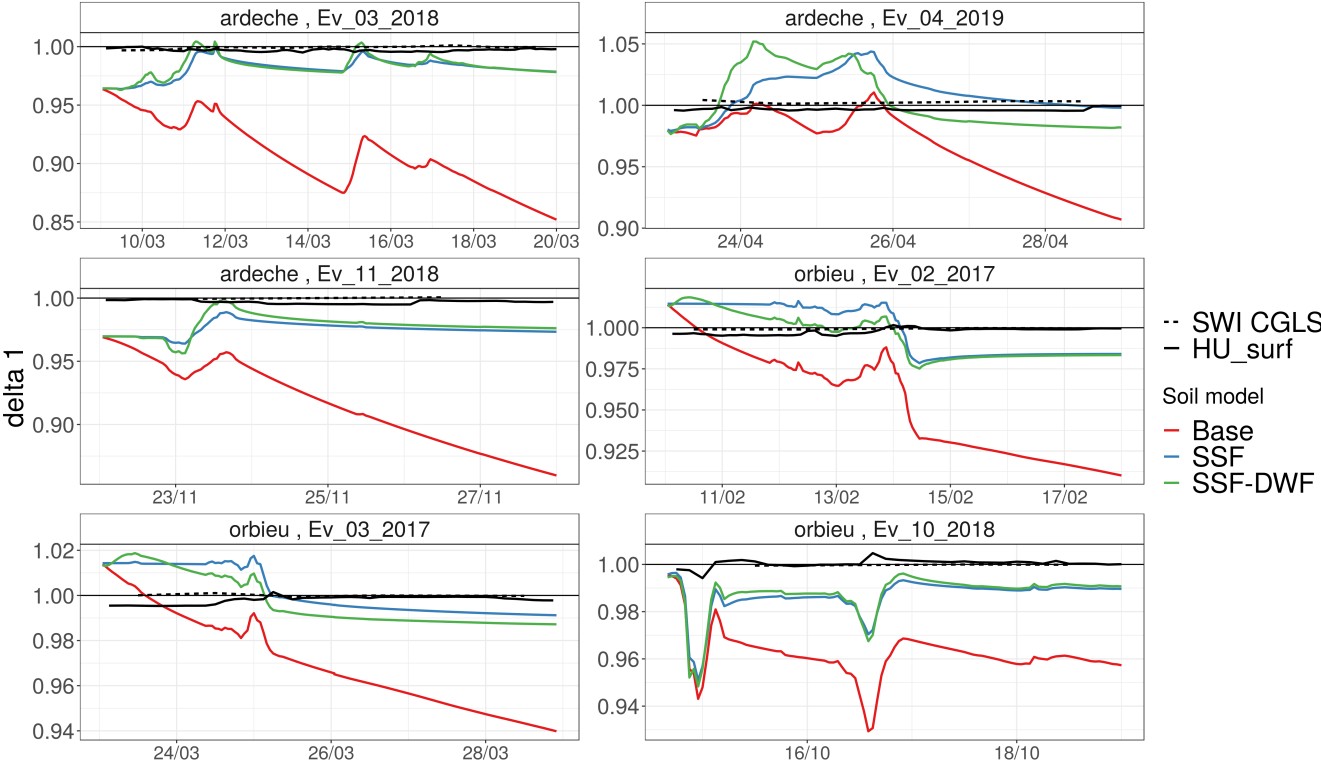

**Figure 9.** Time series of index $\delta 1$ defined by Zoccatelli et al. (2011) for the six events, computed for the soil moisture outputs for the BM, SSF and SSF-DWF models, and also for the LDAS-Monde $HU_{surf}$ variable and the CGLS SWI.

lateral hydraulic conductivities in the deep layer. In this work, the vertical and lateral hydraulic conductivities of the deep layer are considered to be equal. Additional research regarding the deep layer calibration should be led.

For the Orbieu catchment, the extend of the water table (10 $km^2$) is smaller than for the Ardeche catchment, and the response of the piezometric level to precipitation is noticeable. However, its response strongly differs between the three studied events, depending on both the initial piezometric level and the amount of precipitation. For the strong event of October 2018, which started at with a low piezometric level, an increasing of the piezometry is observed immediately with the precipitation, whereas, for the small event of February 2017, the response of the water table is delayed of about two days. None of these

behaviors are represented, neither in the SSF-DWF output, nor in the LDAS-Monde $HU_{deep}$ product.

For the Ardeche catchment, the good correlations between the LDAS-Monde $HU_{deep}$ and the deep layer moisture simulated with the SSF-DWF model highlights the consistency of this model for this catchment, and it corroborates the results of Douinot et al. (2018) which tend to show that this model is particularly suitable for discharge simulation in shale watershed.



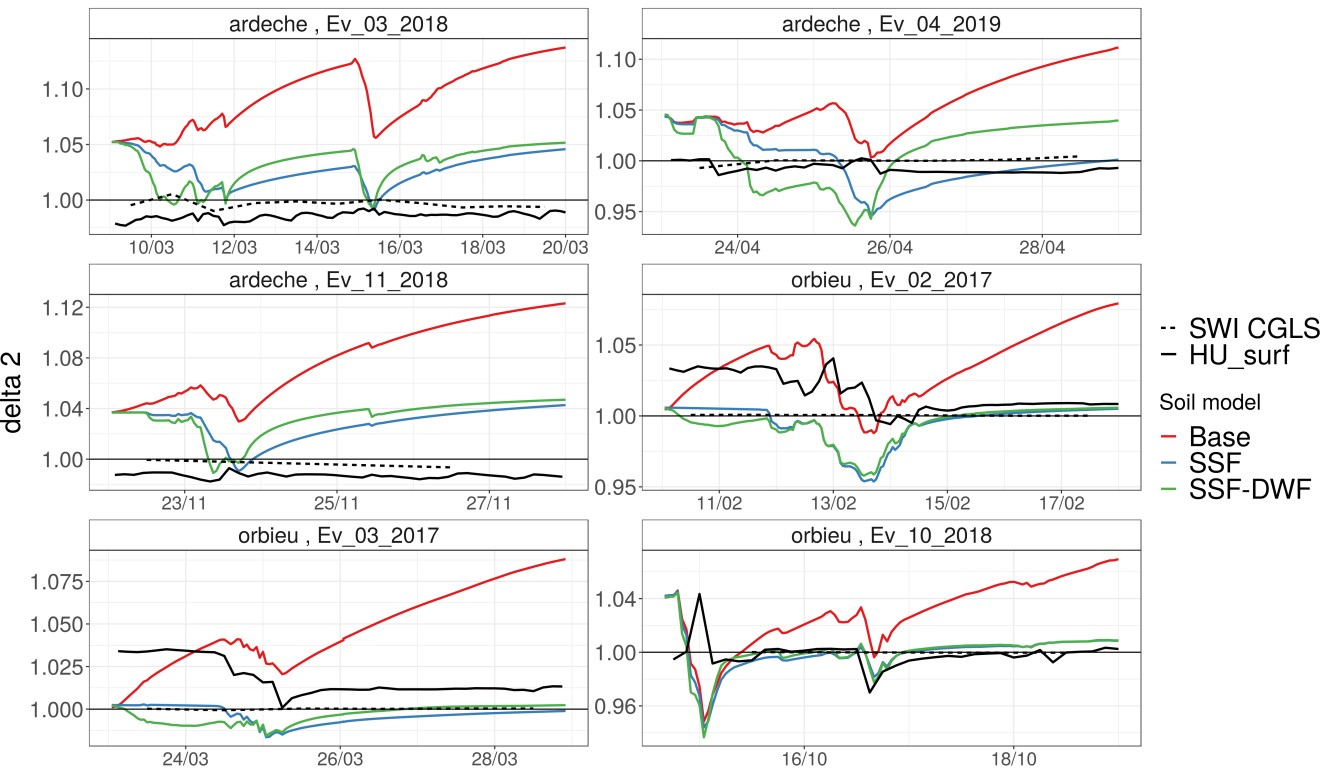

**Figure 10.** Time series of index $\delta2$ defined by Zoccatelli et al. (2011) for the six events, computed for the soil moisture outputs for the BM, SSF and SSF-DWF models, and also for the LDAS-Monde $HU_{surf}$ variable and the CGLS SWI.

Conversely, for the Orbieu catchment, the weak correlations between the LDAS-Monde $HU_{deep}$ and the SSF-DWF model output corroborates the fact that this model seems less well suited for sedimentary catchments. These results illustrate the difficulty to represent the hydrological dynamic of the deep soil layers, with limitation due to the lack of knowledge concerning the physical description of the subsurface water storage (Martin et al., 2004; Maréchal et al., 2013; Vannier et al., 2016).

## 5   Conclusions

The developments of the MARINE model presented by Douinot (2016) are exploited in this work. On the one hand, the transfers through the subsurface are computed based on the volumetric soil water content instead of the water height (SSF model). On the other hand, the soil column is divided into two layers, which represent respectively the upper soil layer and the deep weathered rocks (SSF-DWF model). These developments enhance the degree of refinement of the soil physics described in the model. The impacts of this representation of the subsurface on the water discharge are extensively studied by Douinot (2016).

However, their influence on the spatial dynamic of soil saturation has not yet been explored. This paper aims to assess the



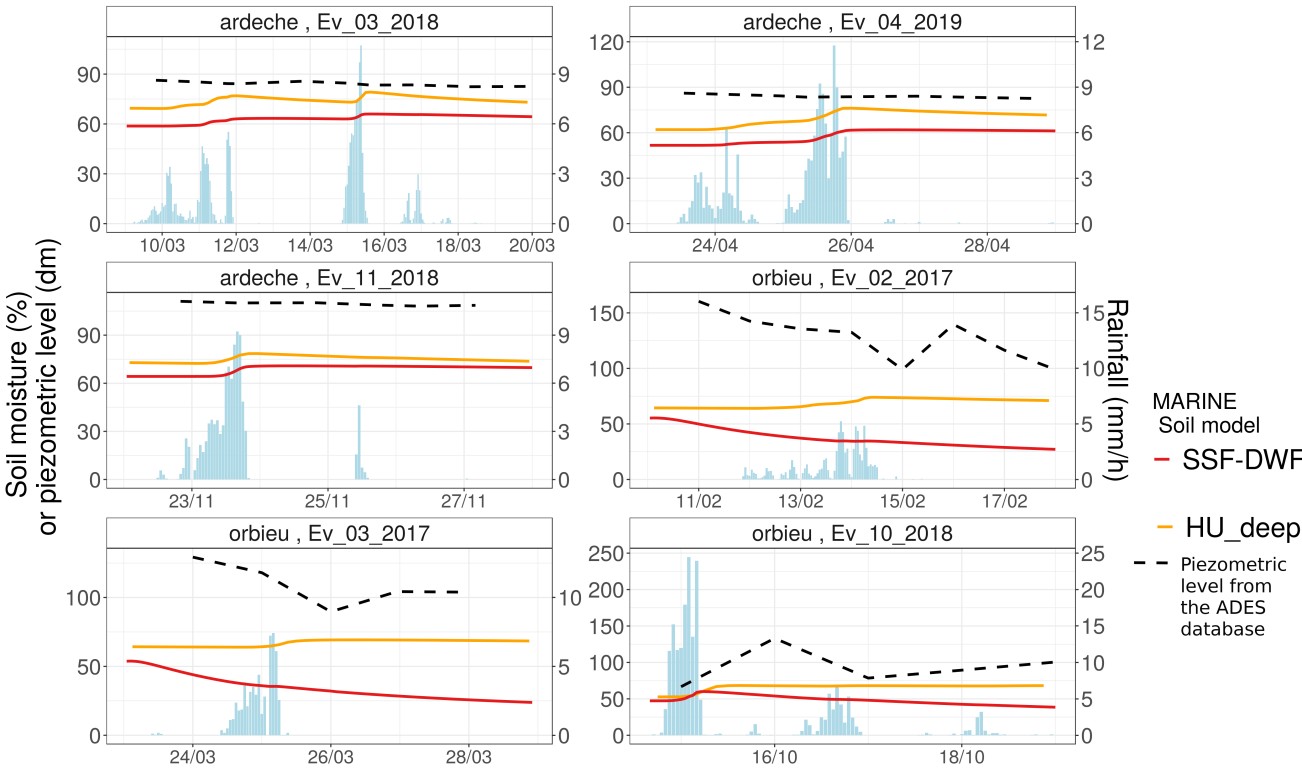

**Figure 11.** Soil moisture simulated for the deep layer with the SSF-DWF model, together with the LDAS-Monde $HU_{deep}$ time series, on average per catchment.

performances of these developments for the representation of soil saturation during flash flood events.

The performances of the model are estimated with respect to several soil moisture products, either at the local scale or spatially extended: i) The gridded soil moisture product provided by the operational modeling chain SAFRAN-ISBA-MODCOU
at the daily time step and at the 8-km resolution; ii) The gridded soil moisture product provided by the LDAS-Monde assimilation chain, based on the ISBA-a-gs land surface model and assimilating high resolution spatial remote sensing data, available at the hourly time step and at the 2.5-km resolution; iii) the upper soil moisture hourly measurements taken from the SMOS-MANIA observation network; iv) The Soil Water Index provided by the Copernicus Global Land Service (CGLS), derived from Sentinel1/C-band SAR and ASCAT satellite data, at the daily time step and at the kilometric resolution. A comparative
assessment of the various products based on remote imagery available for soil moisture in the literature is performed. This literature exploration of the data available for soil moisture description illustrates the difficulty to estimate surface soil moisture based on satellite data at small catchment scale ($\sim 100 km^2$). Considering its satisfying data availability and its fine spatial resolution, the SWI product provided by CGLS is compared with the soil moisture simulated in MARINE. These products





represent valuable indicators of the spatio-temporal dynamics of soil moisture at various scales.


The case study is performed over two catchments located in the South of France, namely the Orbieu river catchment at the Lagrasse station and the Ardeche river catchment at the Vogue station, particularly impacted by flash flood mediterranean events. The study focuses on three flash flood events for each catchment, that occurred between February 2017 and April 2019. These six events present various characteristics, regarding mainly the structures of the pluviometric events and the soil mois-

ture antecedent conditions. The MARINE flash flood model is set up following the calibrations provided by Garambois et al. (2015) for the Orbieu catchment and by Douinot (2016) for the Ardeche catchment. The ANTILOPE QPE data are used as hourly precipitation input for the MARINE model at the kilometric resolution. As the scope of this work is to assess the soil moisture simulation according to the physic considered in the soil models, the discharges simulated with the different models are considered as it is, and the calibrations are not further optimized. The comparison between the gridded soil moisture es-

timates and the local measurements of soil moisture provided by the SMOSMANIA network is performed through a spatial averaging of the gridded simulated values over a $1km^2$ area around the measurement point. As the LDAS-Monde provides soil moisture values for 11 soil layers, these values are synthesized by three summary variables representing respectively the upper soil layer, the deep soil layer and the total soil column. The spatial distributions of soil moisture grids are quantitatively described through the definition of the spatial moments $\delta_1$ and $\delta_2$.


The local comparison of the MARINE outputs for surface soil moisture with the SMOSMANIA measurements, as well as the comparison at the basin scale with the gridded LDAS-Monde and CGLS data lead to the same conclusions: soil moisture simulated with the base model significantly differs from the simulations using the SSF and the SSF-DWF models. When no precipitation happens, the soil layer empties faster with the base model, leading to a simulated soil moisture significantly lower

with the base model than with the two other models. This behavior can be physically explained by the fact that, in the SSF and the SFF-DWF models, the lateral transfers are computed as a function of the volumic soil water gradients, whereas in the base model, they are computed as a function of the water height gradient. Indeed, since the water height gradient between two cells depends on the slope between the cells and the cells textures, water height gradients are larger than volumic soil water gradient when no precipitation happens. Consequently, lateral flows based on the water height gradients are larger than lateral flows

based on the volumic soil water gradient. In addition, the dynamics as well as the amplitudes of the soil moisture simulated in the SSF model and for the upper layer in the SSF-DWF model are better correlated with both the SMOSMANIA measurements and the LDAS-Monde data than the outputs of the base model. Considering that the dynamics of the LDAS-Monde $HU_{surf}$ is of satisfying accuracy, this assessment leads to the conclusion that the SSF-DWF model improves the simulation of the dynamics of the surface layer moisture, compared to both the SSF and the base models. This results appears to be particularly

reliable, since it is observed both a the point measurement scale and at the catchment scale.

In the SSF-DWF model, the simulation of the moisture in the deep layer is also compared to LDAS-Monde moisture data provided for deeper layers, as well as local piezometric measurements available for each catchment. However, the simulation





of the deep layer water content strongly depends on the calibration of the deep layer thickness, the deep layer porosity and the

vertical and lateral hydraulic conductivities in the deep layer. These results illustrate the difficulty to represent the hydrological dynamic of the deep soil layers, with limitation due to the lack of knowledge concerning the physical description of the subsurface water storage. Further conclusions concerning the simulation of deep soil moisture would then require an extensive work to enhance the parametrization of the deep layer in the SSF-DWF model. In particular, the Height Above Nearest Drainage (HAND) method (Nobre et al., 2011) would offer the opportunity to take into account the terrain physical characteristics in the

deep layer parametrization.

In conclusion, this work exposes that enhancing the degree of refinement of the soil physics for the representation of subsurface flow in the MARINE model appears to enhance the upper soil moisture simulation during flash floods, with respect to both spatialized model outputs and satellite-based data, as well as with respect to local soil moisture measurements.

*Author contributions.* JE performed the model simulations and the comparison of the different products, and prepared the paper. HR supervised the work. BB and CA provided the LDAS-Monde product and fed the discussion. AD designed and implemented the SSF and the SSF-DWF models. All authors discussed the results and contributed to the text.

*Competing interests.* The authors declare that they have no conflict of interest.

*Acknowledgements.* This work was funded by the STAE-IRT Saint Exupery fundation, in the framework of the POMME-V project. The
THEAI-Land VHSR data are broadcast through the prism platform, available at www.theia-land.fr. The discharge measurements and the SIM soil moisture data were provided by the French Service for Flood Prevision (SCHAPI). The precipitation data were provided by MeteoFrance. The authors would like to thanks J.P. Wigneron, M. Zribi and N. Baghdadi for providing the SMOS-IC and THEAI-Land VHSR data, respectively, and feeding the discussion on these products.





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
