# Peer review of "A multi-sourced assessment of the spatio-temporal dynamics of soil moisture in the MARINE flash flood model"

_Hydrology and Earth System Sciences, 2020_

## Referee Comment (RC1) · Anonymous Referee #1 · 14 Aug 2020

**OVERVIEW**

The study investigates different formulations of the MARINE flash flood model to assess their performance in reproducing spatial and temporal dynamic of soil moisture during flash flood events. Specifically, three different formulations are compared for 6 flash flood events in two basins. As benchmark, in situ, satellite, and modelled soil moisture observations (plus piezometric head) are used.

**GENERAL COMMENTS**

The paper is fairly well written and clear, but in my opinion several parts need to be improved and other parts corrected. The topic of the paper is interesting for the read-

ership of HESS as the assessment of flash flood modelling tools through soil moisture observations is relevant to understand their reliability and robustness. Therefore, I believe the paper might deserve to be published after addressing the comments I have listed below, with the indication of their relevance.

1) MAJOR: The main result of the paper is that the new formulation of MARINE model (SSF-DWF) is performing better than the base model in terms of reproducing soil moisture dynamic (and river discharge). However, I am not sure that the paper clearly demonstrate this point. The main question is: are the better results related to the new model formulation or to its parameterization? I mean, if the base model is recalibrated I guess it will be able to reproduce soil moisture dynamic as well as the SSF-DWF model. Is that true? This point should be assessed carefully in the paper.

2) MAJOR: I am fully aware of the difficulties in obtaining river discharge observations during flash flood events. However, I believe that 3 flood events per catchment is not enough for a robust assessment. A larger number of events should be assessed, also by selecting smaller events (at least 10-15 events are needed). Otherwise the obtained statistics are too weak to provide robust results.

3) MODERATE: The assessment of deep layer soil moisture through groundwater observations is misleading. Due to the short time periods considered, and the long-term characteristic of groundwater response, the assessment does not provide meaningful results. If the authors do not extend the time period of the analysis, I would suggest to remove this part.

4) MAJOR: Model performance in reproducing river discharge is not good for several events (NSE<0). I am aware that the main objective is the model assessment through soil moisture observations, but if the model is not good in reproducing river discharge I would expect the same with soil moisture. Is it possible to recalibrate the model for such events (and better for a larger number of events) to assess if improving discharge simulations also a benefit in soil moisture reproduction is observed? Otherwise I am

not sure if the model is a suitable tool for simulating soil moisture and river discharge in the selected catchments.

5) MODERATE: The assessment in terms of soil moisture should be carried out only in terms of temporal dynamics. The assessment in terms of absolute values or in terms of range of values is meaningless as the different soil moisture observations have different representativeness in terms of spatial scale and soil depth. Sometimes in the paper it reads this kind of assessment that should be removed.

6) MODERATE: Related to the point above, I would strongly suggest to extend the analysis of spatial patterns. The model capability in reproducing spatial soil moisture patterns is largely unexplored in the scientific literature even though it is a highly relevant topic.

7) MODERATE: I have found the paper too long and difficult to follow in some parts. I would suggest reducing some parts and/or moving them to the appendix. For instance, the analysis of the spatial moments (Figures 9 and 10) does not add important findings to the paper and can be moved to the appendix (or removed). As always in scientific papers, it is better to show a more limited number of figures and tables but more focused to the main message the authors want to convey to the readership.

In the specific comments I have added several suggestions to improve the manuscript (in my opinion). Please address the comments carefully as several parts need to be corrected.

**SPECIFIC COMMENT (L: line or lines)**

L29-35: Several mechanisms of runoff generation do exist, such as infiltration excess, saturation excess, subsurface and deep groundwater flow, flow through macropores and preferential flow. The description in this paragraph is too simplistic and it should be improved.

L45-47: Several studies have demonstrated that local soil moisture measurements are

representative of larger areas and hence they can be useful for initializing flood models (e.g., Brocca et al., 2009 JHE; Tramblay et al., 2010 JoH). Therefore, this part should be partly changed.

L49: I would change "continuous models" with "land surface and distributed hydrological models".

L53-54: The sentence "However, remote sensors ... of surfaces" is not clear and it should be revised. Note that different remote sensing techniques have been developed for obtaining soil moisture from satellite measurements.

L59: Note that also simplified approaches, e.g., Soil Water Index (used also in the paper), have been developed for obtaining root zone soil moisture. They should be mentioned here.

L64: I would change "tested" with "used".

L70-80: Different models and products are mentioned here without references, they should be added.

L82: Change with "and the flood events considered for this study".

L103: Change "volumic" with "volumetric" throughout the text.

L108: Change with "...are defined in the so-called deep water ...".

L116: A figure showing the three different schemes of the MARINE model would help the reader to understand the differences in the model representation.

L119: What does it mean that "the flows in deep layer remains a function of the water height"? Which water height? Is it the water depth in the soil layer? Please clarify.

L126: Change with "particularly prone to flash flood events".

L143-144: An average soil depth of 27 cm and 37 cm for the two catchments seem very thin. Is that correct? What does this parameter represent? I believe that the actual soil

depth is much larger.

L151: Change "pluviometers" with "raingauges", krigging with one "g".

L152: Change with "are available at hourly time step and 1 km resolution".

L152: What are "critized observed discharges"?

L166-167" What are "meteorological antecedents"?

L167: Six events are not enough to guarantee robust results.

Table 2: The uncertainty values are quite strange, I would suggest removing them. It is very hard to provide good numbers as the uncertainty of different products is dependent on many factors.

CGLS SWI should be referred to Bauer. . . et al., 2018b).

ESA CCI is obtained from a number of active and passive sensors, please revise.

L218: ESA CCI should be referenced by Dorigo et al. (2017 RSE doi:10.1016/j.rse.2017.07.001).

L223-224: Figure 2 is not showing the fraction of missing values, please check and revise.

Table 4: Acronyms (BM, SSF, DWF) should be defined in the captions, or a list of acronyms should be provided.

Figure 5: For some events it is evident that poor model performances are due to wrong initialization. How is the model initialized? If the initial soil moisture condition is calibrated, does the model work correctly? This kind of assessment should be carried out. Again, otherwise the model is not a good tool for flash flood prediction (e.g., event March 2017, Orbieu).

Figure 6: Crowded figure, difficult to distinguish the different lines.

L393: Should be "March 2018"?

L405: "to BE consistent"

L441: Kendal correlations of 6.4 and 8.7? Maximum value should be 1.

L442-444: The sentence is not clear and it should be revised.

L478-504: There's no need to repeat in the conclusions the analyses made, remove this part.

**RECOMMENDATION**

Based on the above comments, I suggest a major revision before the possible publication on Hydrology and Earth System Sciences.

---

## Referee Comment (RC2) · Anonymous Referee #2 · 25 Aug 2020

General: This study presents an assessment of two new concepts included in the MARINE flash flood model to increase the schematization of the subsurface. In situ, satellite and model soil moisture estimates are used to validate the model results for flash flood events. Both the temporal as spatial dynamics of soil moisture are studied.

The study presents interesting developments of the MARINE model. Also the comparison with both in situ and innovative satellite soil moisture data leads to interesting insights. The methodology and results are clear and generally well presented. The structure of the manuscript is good, although the conciseness of the manuscript can be improved. Furthermore, the manuscript contains many spelling errors. Based on my

comments, I suggest a major revision of the manuscript. My comments are discussed in the next sections.

Major comments: - The manuscript contains many grammar and spelling errors, which makes the manuscript rather difficult to read. The authors should correct these errors. A (rather long, but not exhaustive) list with proposed technical changes is appended at the end of the review. Also, some parts, especially in the introduction and results sections, need restructuring, as some statements are repeated quite often.

- The authors use only one in situ soil moisture station per catchment for the study. Is this sufficient? Several studies show that the use of only one point location for the validation of gridded soil moisture products introduce large uncertainties (e.g. https://doi.org/10.1016/j.rse.2020.111806, https://doi.org/10.1146/annurev.earth.30.091201.140434 or https://doi.org/10.1029/2011RG000372). Is it possible to compare the absolute values of the different soil moisture datasets while not investigating the same spatial scale? The authors should clearly state the spatial differences between the point observations, model output, and satellite estimates. Also, the authors should explain how these difference in spatial scales affect the findings of the study.

- The authors show an extensive analysis of various datasets. However, some analyses can be investigated more in-depth. As an example: P13, l308-309: "Despite being initially defined by Zoccatelli et al. (2011) to characterize rainfall fields, the delta_1 and delta_2 moments also appear to be particularly relevant when applied to soil moisture fields." –> Please explain how you calculate these moments and why they are relevant when applied to soil moisture fields. What is the consequence on the findings of the delta_1 and delta_2 moments? What do these results mean in context of the model and soil moisture products? I would like to see a discussion included in the manuscript. The same holds for the findings on the spatial variation in soil moisture.

Specific comments: -Title: either use dynamics or variability instead of dynamic -

[Figure]

Abstract, l17-18: "The opportunity of improving the two-layers model calibration is then discussed." Please provide a summary of the discussion instead of referring to the discussion. - Reference section: please provide doi for each reference if available. - The introduction can be improved by including a concise discussion on the relation between soil moisture content and flash floods. - Can you add a short section to the introduction on flash floods and why it is important to model them in France? - Can you be more specific about assessing the performance? What exactly do you mean with performance? Accuracy of model output? Model efficiency? - The authors often refer to "spatially extended data". Consider rephrasing this to "spatially distributed data". - The authors should be more clear on the use of the word soil moisture. An example is shown on page 11, line 275: "the MARINE soil moisture is compared to the moisture of the surface layer". Please indicate whether volumetric soil moisture content or soil saturation degree is considered. - Please move references in the middle of a sentence to the end of that sentence. As an example (P8, l 184): "LDAS-Monde (Albergel et al., 2017) assimilates satellite derived data into the ISBA land surface model." –> LDAS-Monde assimilates satellite derived data into the ISBA land surface model (Albergel et al., 2017). - P2, l25: Please define what an integrative discharge variable is. - P2, l34: "controlling coefficients". Do you refer to the parameters of the discussed representations of infiltration? - P2, l36-37: "This variety.. ..during flood events" This sentence can be removed for clarity. - P2, l42: "the lack of underground flow measurements" –> I believe you want to refer to soil moisture measurements rather than underground flow measurements. Is that correct? - P2, l44: please define event-based hydrological models. - P2, l45-47: I do not fully agree with this statement. If you simulate soil moisture using a 1D-soil column model, point measurements provide valuable information. Could you clarify this sentence? - P2, l48: "continuous models" –> what do you mean with a continuous model in this context? - P2, l50-53: please clarify why using these products lead to structural model uncertainties. - P2, l53-61. This part needs support of references (use of data-assimilation to improve RZSM representation. Examples are: https://doi.org/10.1016/j.jhydrol.2014.08.008,

https://doi.org/10.1016/j.hydroa.2019.100040, and https://doi.org/10.1109/IGARSS.2009.5418264. - P3, l63: Isn't MARINE an abbreviation of Model of Anticipation of Runoff and INundations for Extreme events? - P3: can you include more details on the recent developments of the MARINE model in the introduction? Briefly explain how the representation of subsurface flow was improved? Also, please briefly discuss the foundings of Douinot (2016) after line 68. - P3, l76: "upper soil moisture hourly measurements" –> what's the meaning of upper in this context? - P3, l77: "kilometric resolution" this is a bit ambiguous, please use same definition of spatial resolution as used in lines 73 and 75. - The relationship between the various model components was not directly clear to me. You might consider to add a figure in section 2.1 showing the connections between the model components. - Although the structure of the manuscript is good, the authors provide a lot of information on models and datasets. The readability of the manuscript would greatly improve when a figure showing the research methodology and a table summarizing all data and models used in the study would be added. - P4, l116: What exactly are the hypotheses here? Do you mean the model developments? I would not consider that a hypothesis. - P5, l131-135: Do you have any references for these datasets? - P5, l141, p6, l153: please add reference to reference list. - P5, l143-145: How did you define these depths? Also, what exactly do you mean with a shallow depth (for Orbieu)? - P6, l152: What is the context of critized in this sentence? - P6, l158-159: Can you discuss why the response of discharge to precipitation during this flood was so fast? - It was not clear to me why you introduce the various soil moisture products in section 2.3. Please clarify at the start of this section why you need these products. - P7, l174: what is the depth of the root layer in SIM? - P8, line 184: Is LDAS-Monde a data-assimilation framework? Please make this clear. - P9, line 214: SMOS remote data –> please clarify that these data are obtained using the SMOS satellite. - Figure 2 is not clear to me. The labels are too small and not in English. Please make a clear distinction between SSM and SWI. Also, the difference between 1km and 25km resolution is not clear. In addition, the precipitation data is not visible, consider adding them in a subplot. Furthermore,

according to the manuscript, the figure also shows the respective fraction of missing values. I do not see this in the figure. - P9, l233: How do you define the very local scale? - P11, l256: the authors state that the ADES locations are situated in the study area. However, according to figure 1, the ADES stations are situated outside the catchments. Why do you consider these stations? - P11, l257-258: What is meant with "the water table is 110 km2 large"? Is this the size of that specific aquifer? If so, what is the relation with the in situ soil moisture station? According to figure 1, the groundwater and soil moisture stations are located quite far away from each other. - P11, l265-266: Please describe the eleven soil layers of the LDAS-Monde product in the data section. - P11, l275: "is compared to the moisture of the surface layer" –> Do you mean compared to the surface layer of LDAS-Monde? - P12, l286-289: I don't understand your argumentation here. Why would you consider the drainage network in averaging of a grid/mesh? Also, do you have 16 different grids for the analysis and you choose to exclude 4 of them? Or are you referring to individual cells of the grid? Could you rephrase and clarify? - P13, l291: Add reference for NS-efficiency criterion. Also, is LNP index an abbreviation? Furthermore, please add the units of each term in equation 1. - P13, l317-318: Why do you consider two grids with different spatial resolution? Can you discuss the impact on your results? Also, why do you have a computational time step of 5 minutes while the precipitation input data has a hourly time step? Please explain. - Section 3.2.2: Shouldn't this section be part of the results section? - Table 4: maybe a figure would visualize these data better, or add the numbers in figure 5. - P15, l334-336: I don't understand the argumentation here. Is your argument here that you use the same parametrization for the BM, SSF, and SSF-DWF model variants? However, the SSF and SSF-DWF model variants contains more parameters, so how do you cope with that? - Although you state otherwise on p19, l405-407, the initialization grid shown in Figure 8 is still visible in the flood rising stages of the SSF and SSF-DWF models. Is the model able to reach an equilibrium after initialization or is the model still in a spin-up stage? Did you had a look at this? - P20, l432-433: "The computation of spatial moments for the CGLS SWI might not

lead to robust conclusions." Ok, so what is the consequence for your message? Why even considering these data in the manuscript then? - P23, l454: "Additional research regarding the deep layer calibration should be led." Please rephrase and explain why additional research should be performed. - Figure 11: I am not convinced that you can use the piezometric observations for validation of the deep layer of the SSF-DWF model. (1) please explain why you validate soil moisture simulations with groundwater observations. Or do you investigate groundwater simulations here?(2) the groundwater observations are located outside the study area. Are they valid to use? - Conclusions: please remove references from conclusion section. - Conclusions: too much information is provided in the conclusion section. Please make the section more concise and to the point. - P27, l529: the HAND method is introduced in the conclusions, but not discussed in the rest of the manuscript. Either discuss the HAND method in the discussion or remove this line from the conclusions. - P27, l531-534: "In conclusion, this work exposes that enhancing the degree of refinement of the soil physics for the representation of subsurface flow in the MARINE model appears to enhance the upper soil moisture simulation during flash floods, with respect to both spatialized model outputs and satellite-based data, as well as with respect to local soil moisture measurements." –> This final statement is difficult to read and follow, although I believe this sentence is strong in summarizing the entire manuscript. The authors should clearly rephrase this sentence. - The authors refer to several articles written in French, such as PhD theses. Is the work of these theses not published in English journal articles or other works written in English?

Technical comments: - Abstract, l5: performances –> performance - Abstract: Do you provide the same conclusion twice in l13-19? Please clarify. - P1, l22-24: In my opinion, these sentences can be split for clarity: Extreme precipitation events are expected to increase both in frequency and amplitude in the context of a changing climate (IPCC, 2014). The performances of modelling tools available for operational purposes are of increasing stake. - P2, l25: remove the word "itself". - P2, l27: "models" change to "hydrological models" - P2, l29: "a large panel of formalism" –> please

rephrase. - P2, l34: replace ""whether" by "either" - P2, l48: "consists in" –> "consists of" - P2, l40-41: please rephrase into something like: "They show that uncertainties in the representation of infiltration processes strongly impact both discharge and surface runoff simulations during flood events." - P2, l41-42: please rephrase. - P2, l49: "necessarily biased" –> "inherently biased" - P2, l50: change "by structural uncertainties of the model and uncertainties on model input" to "due to structural uncertainties of the model schematization and model input" - P2, l52: "plateform" –> "platform" - P3, l69: "dynamic" –> either "dynamics" or variability"" - P3, l81-86: please have a look at the use of language in this section and rephrase. - Several occurrences: "physically based" –> "physically-based" - P4, l92: either Darcy's law or the Darcy law. - P4, l95: "saturation hydraulic conductivity" –> "saturated hydraulic conductivity" - P4, l113, p5, l134, p13, l314 and l316, p17, l359: "data base" –> "database" - P5, l121: "model's" –> "model" - P5, l129-130: Awkward English, please rephrase. - P5, l139-140: "in particular with" –> "for example" - P5, l142: "consists in" –> "is situated in" - P5, l147: "found is" –> "found in" - P6, l149: "quantitatives" –> "quantitative" - P6, l154: "event" –> "events" - P6, l161: "Mars" –> "March" - P6, l162: "serie" –> "series" - P6, l162: "have mainly been" –> "mainly have been" - P7, l164-165: please rephrase. - P9, l208: "neural networks" –> "neural network" - P9, l209: "globally covered" –> do you mean approximately? - P9, l209-210: please rephrase. - P9, l228: "this products intercomparison" –> "comparison" - P9, l228: "the products temporal dynamics" –> "the temporal dynamics of the products" - P9, l231: "product" –> "products" - P9, l235: Please rephrase: "the product that offered the most important data availability". - P9, l237: I do not understand the use of the word important in this sentence. Please rephrase. - P10, l 243: "scale" –> "scales" - P11, l256: "One point of measurement" –> "One measurement location" - P11, l259: "water level" –> "groundwater level" - P11, l266-269: "Two behaviors can be distinguished for the different layers: for the five superficial layers, a fast-responding soil moisture and a more stable soil moisture, with a slower response to precipitation and narrower amplitude range for the deeper layers." –> Awkward English, please rephrase. - P12, l280-281: "However, it raises the issue

to compare point measurements to the gridded simulated soil moisture" –> "However, scale differences exist between the point measurements and the gridded simulated soil moisture content/percentage." - P13, l291: "are estimated" –> "is estimated" - P13, l304-308: "The closer of 1 are the delta_1 values, the more centred around the centroid of the catchment is the field. Values of 1 lower that 1 mean that the field get closer from the outlet, whereas values higher that 1 characterize a field overally located on the highest areas of the catchment. The closer of 1 are the 2 values, the more uniform is the distribution of the field. Values of 2 lower that 1 represent an unimodal distribution and values of 2 higher that 1 mode likely represent a multimodal distribution." –> Awkward English, please rephrase. - P15, l332: "However, in the SSF-DWF model, this dynamics is influenced by the contribution of the deep layer, itself mainly controlled by the parametrization of the thickness of this deep layer." Please rephrase. - P16, l346: "on figure 6" –> "in figure 6". - P16, l351-352: "leading to a simulated soil moisture significantly lower with the base model than with the two other models." –> "leading to a simulated soil moisture significantly lower than the SSF and SSF-DWF models." - P17, l358-359: "In addition, the soil moisture simulated for the surface layer with the SSF-DWF is globally higher than for the two other models." –> "In addition, the soil moisture output of the SSF-DWF model are generally larger than the output of the base and SSF models." - P17, l360: What is the context of superior here? - P17, l366: "This shows that the dynamic of the LDAS-Monde HUsurf variable is locally significant with in-situ surface soil moisture measurement. The reliability of the LDAS-Monde HUsurf dynamic for surface soil moisture description can thus be considered as satisfying." –> Please rephrase and revise the word dynamic in the entire manuscript. - P17, l382: "appear" –> "appears" - P17, l383: "both a the point" –> "both at the point" - P18, l390: "On overall" –> "Overall - P18, l394: "is very consistent" –> "is consistent" - P18, l392: "sharp decreases" –> "sharp decrease" - P18, l393-394: "the dynamic of the CGLS SWI is very consistent" –> "the dynamics of the CGLS SWI are very consistent" This error occurs often in the manuscript. Please revise. - P20, l414: "Indeed, precipitation that waters the catchment are doomed to flow toward the outlet." –> Please rephrase.

- P20, l428: "spatial resolution is the LDAS-Monde HUsurf" –> "spatial resolution of the LDAS-Monde HUsurf product" - P20, l430-432: "This can be explained not only by the spatial resolution coarser than for the MARINE outputs but also by the important amount of missing pixel in this data source, in particular for the Ardeche catchment." –> please rephrase. - Please revise the vertical axis label of figure 9. - P22, l450: "emptying of deep soil faster" –> "emptying of deep soil moisture faster" - P26, l518 "is of satisfying accuracy" –> "are of satisfying accuracy"

---

## Author Comment (AC1) · 3 Nov 2020

[hess, manuscript]copernicus

[1]EeckmanJudith [1]RouxHélène [2]DouinotAudrey [3]BonanBertrand [3,4]Albergel-Clément

[1]Institut de Mécanique des Fluides de Toulouse (IMFT), Université de Toulouse, CNRS - Toulouse, FRANCE
[2]Luxembourg Institute of Science and technology, ERIN, Luxembourg
[3]CNRM, Université de Toulouse, Météo-France, CNRS, Toulouse, France

[4]now at European Space Agency Climate Office, ECSAT, Harwell Campus, Didcot, Oxfordshire, UK

Eeckman Judith ju.eeckman@gmail.com

Response to Reviewer N°1

Eeckman et al.

**Response to Reviewer N°1**

November 3, 2020

We wish to thank the referee for his careful evaluation of the manuscript. Please find below the details responses to the comments (in bold). Some modifications of the manuscript are mentioned in italic.

**1) MAJOR: The main result of the paper is that the new formulation of MARINE model (SSF-DWF) is performing better than the base model in terms of reproducing soil moisture dynamic (and river discharge). However, I am not sure that the paper clearly demonstrate this point. The main question is: are the better results related to the new model formulation or to its parameterization? I mean, if the base model is recalibrated I guess it will be able to reproduce soil moisture dynamic as well as the SSF-DWF model. Is that true? This point should be assessed carefully in the paper.**

The model set-up used in the publication is based on the calibration of the base model. As the SSF model doesn't involved additional parameters, the same calibration is used for it. The SSF-DWF model involves another parameter, the deep layer depth

therefore a new calibration is needed which was carried out from the calibration of the base model. Therefore the most tested and probably the most robust calibration is that of the base model. Another argument in favor of this assertion is that the base model has been thoroughly tested over the last ten years or so, and therefore carefully calibrated, including on the catchments studied in this article (Roux et al., 2011 ; Garambois et al., 2013 ; Garambois et al., 2015a ;Garambois et al., 2015b ;Douinot et al., 2018), whereas the SSF-DWF model has just been developed. Thus, if one parameterization is more questionable, it is more that of the SSF-DWF model than that of the base model. Moreover, Douinot et al. (2017) already showed that the base model is not able to reproduce correctly parts of the hydrological response, whatever the parametrization, that's why we argue that the better results are related to the new model formulation.

**2) MAJOR: I am fully aware of the difficulties in obtaining river discharge observations during flash flood events. However, I believe that 3 flood events per catchment is not enough for a robust assessment. A larger number of events should be assessed, also by selecting smaller events (at least 10-15 events are needed). Otherwise the obtained statistics are too weak to provide robust results.**

Additional flood events would indeed by valuable for the study. However, the selection of the flood events is limited by the period of availability of the LDAS-Monde product at fine scale (2.5 kilometer resolution and 3 hours time step), i.e. since July 2017, as quoted in section 2.3.2 (Bonan et al., 2020). During the July 2017- April 2019 period, only 3 flood events occurred on each of the catchments. No additional flood events can be added for this period over the studied catchments as we couldn't make any comparising for them. Nevertheless, we think that the results concerning 6 events

on 2 different catchments can constitute a first interesting analysis because they all go in the same direction: the interest of the SSF-DWF functioning hypothesis for the representation of the saturation.

The following sentence is added in section 2.2.2: 'This period is chosen because it correponds to the period of availability of the LDAS-Monde product'

**3) MODERATE: The assessment of deep layer soil moisture through groundwater observations is misleading. Due to the short time periods considered, and the long-term characteristic of groundwater response, the assessment does not provide meaningful results. If the authors do not extend the time period of the analysis, I would suggest to remove this part.**

The authors agree with this point. The use of the piezometric measurements had been attempted in order to add more validation data source. The reference to the ADES data, as well as the values in the tables and graphics are removed. The ADES network is only quoted in the last paragraph 4.2.3 Water content of the deep layer, as a possible opening for further works.

**4) MAJOR: Model performance in reproducing river discharge is not good for several events (NSE<0). I am aware that the main objective is the model assessment through soil moisture observations, but if the model is not good in reproducing river discharge I would expect the same with soil moisture. Is it possible to recalibrate the model for such events (and better for a larger number of events) to assess if improving discharge simulations also a benefit in soil moisture reproduction is observed? Otherwise I am not sure if the model is a suitable tool for simulating soil moisture and river discharge in the selected catchments.**

Some performances are indeed poor but when the performance is poor for the base model, it is also poor for the SSF and SSF-DWF models, that's why we argue that it doesn't affect the results of the model inter-comparison. Indeed, several other calibration tests could have been carried out so as to improve the results of the hydrological models. However, the main purpose of this study focuses on a model inter-comparison to test several functioning hypothesis and assessed their respective potential for the simulation of soil moisture dynamic in the same context.

**5) MODERATE: The assessment in terms of soil moisture should be carried out only in terms of temporal dynamics. The assessment in terms of absolute values or in terms of range of values is meaningless as the different soil moisture observations have different representativeness in terms of spatial scale and soil depth. Sometimes in the paper it reads this kind of assessment that should be removed.**

The absolute values of the different soil moisture data are not directly compared. A special attention is paid not to do so when comparing different data source. When comparing the MARINE soil moisture outputs from the different models (BM, SSF or SSF-DWF), the soil moisture values are compared in terms of absolute values. In this case, the absolute values are comparables because they represent the same physical variable. Yet, the soil depth considered in each model is specified.

**6) MODERATE: Related to the point above, I would strongly suggest to extend the analysis of spatial patterns. The model capability in reproducing spatial soil**

**moisture patterns is largely unexplored in the scientific literature even though it is a highly relevant topic.**
We definitely agree that the comparison of spatial patterns is of high interest for the improvement of the understanding of hydrological phenomena. As the direct comparison of maps with different spatial resolution is not straighforward, we choose to consider instead the spatial moment values as a proxy for this analysis.

The paragraph 4.2 Comparison at the catchment scale is splitted into 2 paragraphs :
4.2.1 Catchment average behavior and 4.2.2 Spatial variability.
In section 4.2.2, the spatial variability of soil moisture fields, as well as the conclusion drawn from the spatial moments values are values are detailled.

The paragraph detailing the delta1 dynamics is reformulated as :
" *The general behavior of the $\delta 1$ spatial moment when computed on the SSD is that the $\delta 1$ increases when precipitation happens and then decreases at a variable rate. Indeed, as precipitation necessarily flows towards the outlet, $\delta 1$ values are bound to increase (i.e. the SSD fields get closer from the oultet after a precipitation event. The $\delta 1$ time series obtained with both the SSF and the SSF-DWF models are sig nificantly closer to 1 than the $\delta 1$ values obtained with the base model. This means that the SSD fields simulated with the base model are globally closer from the outlet than with the SSF and the SSF-DWF models, that is to say that the propagation of the water throught the drainage network in the upper soil layer is faster for the base model than for the SSF and the SSF-DWF models. The analysis of the $\delta 1$ time series allows to quantify the impact of the calibration of lateral transfers on the SSD distribution*. "

The paragraph detailing the $\delta 2$ dynamics is reformulated as :
" The $\delta 2$ values for the SSF and SSF-DWF models are globally closer to 1 than for the base model, that is to say that the SSD fields simulated with the SSF and SSF-DWF

models are globally more uniform than for with the base model. This can be explained by the fact that the SSD is globally higher for the SSF and SSF-DWF models than for the base model (see figure **??**), the difference between the SSD and saturation in the drainage network (i.e. 100 %) is stronger for the base model than for the other two models. This leads to SSD fields more uniform for the SSF and SSF-DWF models than for the base model. This result is particularly observed for the Orbieu catchment. The analysis of the $\delta 2$ time series allows to quantify the differences between one the one side, base model, and on the other side the SSF and the SSF-DWF models. "

The following synthetic sentence is added at the end of 4.2.2:
" *The analysis of the delta1 and delta2 spatial moments provides an inovative way to assess the spatial variability of the SSD fields. The reaction of the SSD fields to precipitation are quantified. The difference between the spatial repartition of the ouputs of the base model on the one side and the SSF and SSF-DWF models on the other side, is highlighted.* "

**7) MODERATE: I have found the paper too long and difficult to follow in some parts. I would suggest reducing some parts and/or moving them to the appendix. For instance, the analysis of the spatial moments (Figures 9 and 10) does not add important findings to the paper and can be moved to the appendix (or removed). As always in scientific papers, it is better to show a more limited number of figures and tables but more focused to the main message the authors want to convey to the readership. In the specific comments I have added several suggestions to improve the manuscript (in my opinion). Please address the comments carefully as several parts need to be corrected.**

A particular care has been paid to improve the clarity of the paper. The technical

changes pointed below, as well as spelling errors have been corrected. The clarity of the paper have been improved by selecting essential information. A table has been added to present the different data source for soil moisture. A scheme has been added to present the three different versions of the MARINE model (BM, SF and SSF-DWF). The litterature review of the various satellite data source, the detail of the different soil layers in the LDAS-Monde product have been moved to an appendix section. The introduction is completed and reformuled. The conclusion is shortened and reformuled. As explained above, the spatial moments have been chosen to perform the analysis of spatial patterns which is crucial for distributed hydrological modelling. That is why we keep this analysis in the article reformulating it to make it more relevant.

**SPECIFIC COMMENT :**
**L29-35: Several mechanisms of runoff generation do exist, such as infiltration excess, saturation excess, subsurface and deep groundwater flow, flow through macropores and preferential flow. The description in this paragraph is too simplistic and it should be improved.**

A paragraph describing the processes is inserted (see below) and the paragraph on representation of the subsurface in the models is reformulated.

*" Several mechanisms generate the partition between infiltration and surface runoff. Surface runoff can happen when rainfall intensity excess the maximum infiltration rate of the soil (infiltration excess), or when the precipitation volumes exceed the storage capacity of the soil (saturation excess). Then, the generation of surface runoff directly rely on the water content of the subsurface. Within the subsurface, both vertical infiltration flows and lateral transfers take place. These flow are controlled by the*

*physical characteristics of the porous media, such as its hydraulic conductivity or its capacity at saturation. In addition, preferential flows happen through macropores or fractured aquifers. "*

**L45-47: Several studies have demonstrated that local soil moisture measurements are representative of larger areas and hence they can be useful for initializing flood models (e.g., Brocca et al., 2009 JHE; Tramblay et al., 2010 JoH). Therefore, this part should be partly changed.**
The sentence is replaced by :
*" Several studies have demonstrated that local soil moisture measurements are representative of relatively larger areas and hence they can be compared to spatially distributed simulation outputs around the point of measurement (Brocca et al., 2009; Tramblay et al., 2010) "*

**L49: I would change "continuous models" with "land surface and distributed hydrological models".**
The edit has been done.

**L53-54: The sentence "However, remote sensors . . . of surfaces" is not clear and it should be revised. Note that different remote sensing techniques have been developed for obtaining soil moisture from satellite measurements.**
The following sentence is added :
*"Satellite imagery provides valuable spatially distributed data. Different remote sensing techniques have been developed for obtaining soil moisture from satellite measurements. "*

**L59: Note that also simplified approaches, e.g., Soil Water Index (used also in the paper), have been developed for obtaining root zone soil moisture. They should be mentioned here.**
The edit has been done.

**L64: I would change "tested" with "used".**
The edit has been done.

**L70-80: Different models and products are mentioned here without references, they should be added.**
The references are added for each models and data.

**L82: Change with "and the flood events considered for this study".**
The edit has been done.

**L103: Change "volumic" with "volumetric" throughout the text.**
The edit has been done.

**L108: Change with ". . .are defined in the so-called deep water . . .".**
The edit has been done.

**L116: A figure showing the three different schemes of the MARINE model would help the reader to understand the differences in the model representation.**
A figure is added to summarize the main state variables and flux in the soil for the three models.

**L119: What does it mean that "the flows in deep layer remains a function of the water height"? Which water height? Is it the water depth in the soil layer? Please clarify.**
Reformulated as :
*" In the SSF-DWF, the flows in the deep layer is defined as a function of the water height in the deep layer "*

**L126: Change with "particularly prone to flash flood events".**
The edit has been done.

**L143-144: An average soil depth of 27 cm and 37 cm for the two catchments seem very thin. Is that correct? What does this parameter represent? I believe that the actual soil depth is much larger.**
The soil depth values are taken from the INRA databases. These databases have been established for agronomic uses and do not document deep‐weathered rock horizons (i.e. pedologic horizons of type C and deeper) (Vannier et al., 2014). The average soil for France is about 1 m. These values look correct, considering the physiography of the two catchments.

**L151: Change "pluviometers" with "raingauges", krigging with one "g".**
The edit has been done.

**L152: Change with "are available at hourly time step and 1 km resolution".**
The edit has been done.

**L152: What are "critized observed discharges"?**
" critized discharges" are defined as discharge measurements that have been cor-
rected from known biases. In particular, in the case of flood events, high water heights are difficult to extrapolate from the rating curve. Direct measurements have then to be criticized and corrected according to humain expertise. This analysis is carried out by the forecasters of the French national flood forecasting services.

**L166-167" What are "meteorological antecedents"?**
The expression "meteorological antecedents" refers to the paper of Berthet et al.,2009 : " How crucial is it to account for the antecedent moisture conditions in flood forecasting? ". This work shows the impact of the hydro-meteorological situtation during the days before the flood. For clarity purpose, this word is removed from the text.

**L167: Six events are not enough to guarantee robust results.**
Additional flood events would indeed by valuable for the study. However, the selection of the flood events is limited by the period of availability of the LDAS-Monde product at fine scale (2.5 kilometer resolution and 3 hours time step), i.e. since July 2017, as quoted in section 2.3.2 (Bonan et al., 2020). During the July 2017- April 2019 period, only 3 flood events occured on each of the catchments. No addictional flood events can be added for this period over the studied catchments as we couldn't make any comparison for them. Nevertheless, we think that the results concerning 6 events on 2 different catchments can constitute a first interesting analysis because they all go in the same direction: the interest of the SSF-DWF functioning hypothesis for the representation of the saturation.

**Table 2: The uncertainty values are quite strange, I would suggest removing them. It is very hard to provide good numbers as the uncertainty of different products is dependent on many factors.**
The uncertainties values are taken from the respective paper for each product. But indeed, the meaning of the uncertainty value might differ between the studies. The

uncertainty column is then removed from the table.

**CGLS SWI should be referred to Bauer. . . et al., 2018b). ESA CCI is obtained from a number of active and passive sensors, please revise.**
The reference has been corrected.

**L218: ESA CCI should be referenced by Dorigo et al. doi:10.1016/j.rse.2017.07.001).**
The reference has been corrected.

**L223-224: Figure 2 is not showing the fraction of missing values, please check and revise.**
The graphical appearance of this figure is enhanced. The figure is sent to the appendix section .

**Table 4: Acronyms (BM, SSF, DWF) should be defined in the captions, or a list of acronyms should be provided.**
The meaning of the acronyms is added in the caption.

**Figure 5: For some events it is evident that poor model performances are due to wrong initialization. How is the model initialized? If the initial soil moisture condition is calibrated, does the model work correctly? This kind of assessment should be carried out. Again, otherwise the model is not a good tool for flash flood prediction (e.g., event March 2017, Orbieu).**

All the models (base, SSF, SSF-DWF) are initialized with the spatial soil saturation

outputs from from Météo-France's SIM (Safran-Isba-Modcou, Habets et al., 2008) operational chain, the initial soil moisture is not calibrated. The first bump in the hydrograph is due to the hydrological functioning as with the same initialization the SSF model doesn't present the same behaviour. This is therefore an integral part of what this article seeks to analyze. Indeed, as stated above, the main purpose of this study is a model intercomparison to test several functioning hypothesis in the same context.

**Figure 6: Crowded figure, difficult to distinguish the different lines.**
Different shades of grey are used for the SMOSMANIA data lines.

**L393: Should be "March 2018"?**
Yes,corrected.

**L405: "to BE consistent"**
Corrected

**L441: Kendal correlations of 6.4 and 8.7? Maximum value should be 1.**
These are percent values. Corrected in the text

**L442-444: The sentence is not clear and it should be revised.**
This sentence is reformulated as :
*" However, for this catchment, the response of the piezometric level to the precipitation is small for the studied events. This can be explained by the fact that the extend of the water table (1849 km$^2$) is large compared to the Ardeche catchment area (622km$^2$). Consequenlty, the variations of the piezometric level is not reliable to assess the simulated moisture of the deep layer for this catchment during flash flood events. "*

**L478-504: There's no need to repeat in the conclusions the analyses made, remove this part.**
This part is removed.

**References**

Bonan, B., Albergel, C., Napoly, A., Zheng, Y. and Calvet, J.-C., 2020. An offline reanalysis of land surface variables with LDAS-Monde forced by a kilometric scale NWP system. EGU General Assembly 2020. https://doi.org/10.5194/egusphere-egu2020-17527.

Douinot, A., Roux, H. and Dartus, D., 2017. Modelling errors calculation adapted to rainfall – runoff model user expectations and discharge data uncertainties. Environmental Modelling and Software. 90, 157-166. doi : 10.1016/j.envsoft.2017.01.007.

Douinot, A., Roux, H., Garambois, P.-A., and Dartus, D., 2018. Using a multihypothesis framework to improve the understanding of how dynamics during flash floods. Hydrol. Earth Syst. Sci. 22, 5317-5340. doi : 10.5194/hess-22-5317-2018.

Garambois, P. A., Roux, H., Larnier, K., Castaings, W., and Dartus, D., 2013. Characterization of process-oriented hydrologic model behavior with temporal sensitivity analysis for flash floods in Mediterranean catchments. Hydrol. Earth Syst. Sci., 17, 2305-2322, doi :10.5194/hess-17-2305-2013.

Garambois, P.-A., Roux, H., Larnier, K., Labat, D. and Dartus, D., 2015a. Characterization of catchment behaviour and rainfall selection for flash flood dedicated hydrologic model regionalization : catchments of the eastern Pyrenees. Hydrological Sciences Journal. 60(3), 424-447.

Garambois, P.-A., Roux, H., Larnier, K., Labat, D. and Dartus, D., 2015b. Parameter regionalization for a process oriented distributed model dedicated to flash floods. J. Hydrol, 525(0), 383-399. https ://doi.org/10.1016/j.jhydrol.2015.03.052.

Habets, F., Boone, A., Champeaux, J.-L., Etchevers, P., Franchisteguy, L., Leblois, E., Ledoux, E., Le Moigne, P., Martin, E., Morel, S., Noilhan, J., Quintana Seguí, P., Rousset-Regimbeau, F., and Viennot, P.: The SAFRAN-ISBA-MODCOU hydrometeorological model applied over France, J. Geophys. Res.-Atmos., 113, D6, https://doi.org/10.1029/2007JD008548, 2008.

Roux, H., Labat, D., Garambois, P.-A., Maubourguet, M.-M., Chorda, J. and Dartus, D., 2011. A physically-based parsimonious hydrological model for flash floods in Mediterranean catchments. Nat. Hazards Earth Syst. Sci. J1 - NHESS, 11(9), 2567-2582.

Vannier, O., Braud, I. and Anquetin, S. 2014. Regional estimation of catchment-scale soil properties by means of streamflow recession analysis for use in distributed hydrological models. Hydrological Processes, 28(26). https://doi.org/10.1002/hyp.10101.

---

## Author Comment (AC2) · 3 Nov 2020

[hess, manuscript]copernicus

[1]EeckmanJudith [1]RouxHélène [2]DouinotAudrey [3]BonanBertrand [3,4]Albergel-Clément

[1]Institut de Mécanique des Fluides de Toulouse (IMFT), Université de Toulouse, CNRS - Toulouse, FRANCE
[2]Luxembourg Institute of Science and technology, ERIN, Luxembourg
[3]CNRM, Université de Toulouse, Météo-France, CNRS, Toulouse, France

[4]now at European Space Agency Climate Office, ECSAT, Harwell Campus, Didcot, Oxfordshire, UK

Eeckman Judith ju.eeckman@gmail.com

Response to Reviewer N°2

Eeckman et al.

[Figure]

**Response to Reviewer N°2**

November 3, 2020

We wish to thank the referee for his/her careful evaluation of the manuscript as well as its very useful and exhaustive corrections. Please find below the details responses to the comments (in bold). Some modifications of the manuscript are mentioned in italic.

**Major comments:**

**\* The manuscript contains many grammar and spelling errors, which makes the manuscript rather difficult to read. The authors should correct these errors. A (rather long, but not exhaustive) list with proposed technical changes is appended at the end of the review. Also, some parts, especially in the introduction and results sections, need restructuring, as some statements are repeated quite often.**

The technical changes pointed below, as well as spelling errors have been corrected. The clarity of the paper have been improved by selecting essential information. The litterature review of the various satellite data source, the detail of the different soil layers in the LDAS-Monde product have been moved to an appendix section.

The introduction is completed and reformuled. The conclusion is shortened and reformuled.

**\* The authors use only one in situ soil moisture station per catchment for the study. Is this sufficient? Several studies show that the use of only one point location for the validation of gridded soil moisture products introduce large uncertainties.**

Is it possible to compare the absolute values of the different soil moisture datasets while not investigating the same spatial scale? The authors should clearly state the spatial differences between the point observations, model output, and satellite estimates. Also, the authors should explain how these difference in spatial scales affect the findings of the study. The comparison of soil measurements to gridded products obviously raises consistency issues. One of the main concerns of the paper is to address these issues. Additionnal soil moisture points measurement would indeed be valuable. But the SMOSMANIA network is the most dense soil observation network available for the south of France. The studied catchments are chosen because they contain one SMOSMANIA point. As stated by reviewer n°1, even if the uncertainties cannot be fully avoided, several studies have demonstrated that local soil moisture measurements are representative of larger areas and hence they can be useful to assess the temporal dynamic simulated by flood models (e.g., Brocca et al., 2009; Tramblay et al., 2010).

The absolute values of the different soil moisture data are not directly compared. A special attention is paid not to do so when comparing different data source. When comparing the MARINE soil moisture outputs from the different models (BM, SSF or SSF-DWF), the soil moisture values are compared in terms of absolute values. In this case, the absolute values are comparables because they represent the same physical

variable. Yet, the soil depth considered in each model is specified.

A table (Table2), as well as the following lines are added in section 2.3. " Available soil moisture data " in order to clarify the spatial differences between the point observations, model output, and satellite estimates : The table 2 summarizes the five products compared in this work for soil moisture estimation: The SAFRAN-ISBA-MODCOU (SIM) root zone soil moisture, the LDAS-Monde root zone soil moisture, the CGLS Soil Water Index (SWI) and the soil moisture measurements provided by the SMOSMANIA network. For the SIM, LDAS-Monde and SMOSMANIA soil moisture data, the soil saturation degree is retrieve by dividing the soil moisture values by its saturation value in the respective product.

**\* The authors show an extensive analysis of various datasets. However, some analyses can be investigated more in-depth. As an example: P13, l308-309: "Despite being initially defined by Zoccatelli et al. (2011) to characterize rainfall fields, the delta1 and delta2 moments also appear to be particularly relevant when applied to soil moisture fields." > Please explain how you calculate these moments and why they are relevant when applied to soil moisture fields. What is the consequence on the findings of the delta1 and delta2 moments? What do these results mean in context of the model and soil moisture products? I would like to see a discussion included in the manuscript. The same holds for the findings on the spatial variation in soil moisture.**

The spatial moments are initially developed by Zoccatelli et al. (2011) to be applied to precipitation fields. In this paper, we propose to apply them to soil moisture fields. This choice is an innovative way to assess the spatial variation of soil moisture fields and easily compare spatial patterns with different spatial resolutions.

The following lines are added in section 3.2 Indices, when introducing the spatial moments :

*'The exact formulation of the $\delta 1$ and $\delta 2$ spatial moments as functions of the spatially distributed field and of the distance to the river 'network can be found in equation 2 and equation 3 in Zoccatelli et al. (2011)'*

In addition, the paragraph 4.2 Comparison at the catchment scale is splitted into 2 paragraphs :

4.2.1 Catchment average behavior and 4.2.2 Spatial variability.

In section 4.2.2, the spatial variability of soil moisture fields, as well as the conclusion drawn from the spatial moments values are detailed.

The paragraph detaling the $\delta 1$ dynamics is reformulated as :

*" The general behavior of the $\delta 1$ spatial moment when computed on the SSD is that the $\delta 1$ increases when precipitation happens and then decreases at a variable rate. Indeed, as precipitation necessarily flows towards the outlet, $\delta 1$ values are bound to increase (i.e. the SSD fields get closer from the oultet after a precipitation event. The $\delta 1$ time series obtained with both the SSF and the SSF-DWF models are sig nificantly closer to 1 than the $\delta 1$ values obtained with the base model. This means that the SSD fields simulated with the base model are globally closer from the outlet than with the SSF and the SSF-DWF models, that is to say that the propagation of the water throught the drainage network in the upper soil layer is faster for the base model than for the SSF and the SSF-DWF models. The analysis of the $\delta 1$ time series allows to quantify the impact of the calibration of lateral transfers on the SSD distribution."*

The paragraph detailing the $\delta 2$ dynamics is reformulated as :

*" The $\delta2$ values for the SSF and SSF-DWF models are globally closer to 1 than for the base model, that is to say that the SSD fields simulated with the SSF and SSF-DWF models are globally more uniform than for with the base model. This can be explained by the fact that the SSD is globally higher for the SSF and SSF-DWF models than for the base model (see figure ??), the difference between the SSD and saturation in the drainage network (i.e. 100 %) is stronger for the base model than for the other two models. This leads to SSD fields more uniform for the SSF and SSF-DWF models than for the base model. This result is particularly observed for the Orbieu catchment. The analysis of the $\delta2$ time series allows to quantify the differences between one the one side, base model, and on the other side the SSF and the SSF-DWF models. "*

The following synthetic sentence is added at the end of 4.2.2:
*" The analysis of the delta1 and delta2 spatial moments provides an inovative way to assess the spatial variability of the SSD fields. The reaction of the SSD fields to precipitation are quantified. The difference between the spatial repartition of the ouputs of the base model on the one side and the SSF and SSF-DWF models on the other side, is highlighted. "*

**Specific comments:**

 **-Title: either use dynamics or variability instead of dynamic**
'dynamic' is remplaced by " dynamics "

**Abstract, l17-18: "The opportunity of improving the two-layers model calibration is then discussed." Please provide a summary of the discussion instead of**

**referring to the discussion.**
Reformulated as : *' Finally, the soil moisture simulated by the two-layers model for the deep layer is compared to the soil moisture provided by the LDAS-Monde product at corresponding depths. '*

**- Reference section: please provide doi for each reference if available.**
The references are provided under the style required by the editor.

**-The introduction can be improved by including a concise discussion on the relation between soil moisture content and flash floods.**
A paragraph describing the processes is inserted (see below) and the paragraph on representation of the subsurface in the models is reformulated as:
*" Several mechanisms generate the partition between infiltration and surface runoff. Surface runoff can happen when rainfall intensity excess the maximum infiltration rate of the soil (infiltration excess), or when the precipitation volumes exceed the storage capacity of the soil (saturation excess). Then, the generation of surface runoff directly rely on the water content of the subsurface. Within the subsurface, both vertical infiltration flows and lateral transfers take place. These flow are controlled by the physical characteristics of the porous media, such as its hydraulic conductivity or its capacity at saturation. In addition, preferential flows happen through macropores or fractured aquifers. "*

**- Can you add a short section to the introduction on flash floods and why it is important to model them in France?**
Lines 21-29 are dedicated to this purpose
Inserted l24 :
*" In particular, modeling systems for short term predictions represent valuable tool for decision making and organization of emergency systems. "*

**- Can you be more specific about assessing the performance? What exactly do you mean with performance? Accuracy of model output? Model efficiency?**
When refering to the performance of the simulated soil moisture, with respect to reference products (SIM, LDAS-Monde or SMOSMANIA mesurements), 'performance' is replaced by 'efficiency'. When refering to 'model performance', 'performance' is used is a more general way, including simulated discharge and simulated soil moisture accuracy. In this case, 'model performance' is replaced by 'model accuracy'. When reference to 'performance criterias' (NSE, etc..), 'performance' is kept.

**- The authors often refer to "spatially extended data". Consider rephrasing this to "spatially distributed data".**
The occurrences of "spatially extended data" are replaced by "spatially distributed data"

**- The authors should be more clear on the use of the word soil moisture. An example is shown on page 11, line 275: "the MARINE soil moisture is compared to the moisture of the surface layer". Please indicate whether volumetric soil moisture content or soil saturation degree is considered.**

For each data source, the meaning of the word "soil moisture " is detailed in the text :
- P9, L207, for the SIM product : *'The soil saturation degree of the root zone (i.e. the volumetric soil water content divided by its value at saturation) is directly provided by the SCHAPI for this work.'*

- P9, L224, for the LDAS-Monde product : *'LDAS-Monde provide both the soil water content and the maps of soil water content at saturation for each of this 11 layers. For each layer, the soil saturation degree is retrieved by dividing its soil water content by the soil water content at saturation.'*

- P11 L251 for the SMOSMANIA network : *" For each sensor, the soil saturation degree is retrieved by dividing the measured soil water content by its value at saturation estimated at the location of the point of measurement. "*

When refering to soil water content, the word ' soil moisture' is replace by 'soil water content'. When refering to soil saturation degree, it is replaced by 'soil saturation' (SSD). When refering to either soil saturation or soil water content, 'soil moisture' is kept.

**- Please move references in the middle of a sentence to the end of that sentence. As an example (P8, l 184): "LDAS-Monde (Albergel et al., 2017) assimilates satellite derived data into the ISBA land surface model."**
The edits are done

**- P2, l25: Please define what an integrative discharge variable is.**
The sentence is reformulated as : *'.. the discharge variable, that integrates all the processes taking place at the subsurface and the surface of the catchment.'*

**- P2, l34: "controlling coefficients". Do you refer to the parameters of the discussed representations of infiltration?**
Yes. Reformulated as : *" parameters controling the representation of infiltration.. "*

**- P2, l36-37: "This variety.. ..during flood events" This sentence can be removed for clarity.**
The sentence is removed

**- P2, l42: "the lack of underground flow measurements" –> I believe you want to refer to soil moisture measurements rather than underground flow measurements. Is that correct?**

Reformulated as :   " *In addition, both the lack of soil and deep ground description and the uncertainties associated with soil moisture estimations lead to an hazardous validation of the model outputs* "

**- P2, l44: please define event-based hydrological models.**

" event-based models " are set up on short time period, typically a flash flood, and only represent the short-term processes that are not neglectible compared to the intensity of precipitation. In particular, the evapotranspiration is commonly not represented in the event-based models. " event-based " models are opposed to " continuous models ". For clarity purpose, " event-based " is removed from the text.

**- P2, l45-47: I do not fully agree with this statement. If you simulate soil moisture using a 1D-soil column model, point measurements provide valuable information. Could you clarify this sentence?**

This part of the introduction is reformulated :  " *local ground measurements provide locally accurate estimations of soil moisture at shallow depths. Several studies have demonstrated that local soil moisture measurements are representative of relatively larger areas and hence they can be compared to spatially distributed simulation outputs around the point of measurement (Brocca et al., 2009; Tramblay et al., 2010).* "

**- P2, l48: "continuous models" –> what do you mean with a continuous model in this context?**

Replaced by 'land surface model and distributed models '

**- P2, l50-53: please clarify why using these products lead to structural model uncertainties.**
Any model output is necessarily associated with uncertainties due to the assumption and parametrization (necessarily reductive compared to real, complex systems). For clarity purpose, the sentence is removed.

**- P2, l53-61. This part needs support of references (use of data-assimilation to improve RZSM representation.**
The three references are added.

**- P3, l63: Isn't MARINE an abbreviation of Model of Anticipation of Runoff and INundations for Extreme events?**
Yes, the edit is done

**P3: can you include more details on the recent developments of the MARINE model in the introduction? Briefly explain how the representation of subsurface flow was improved? Also, please briefly discuss the foundings of Douinot (2016) after line 68.**

The following lines are added in the introduction :
*" On the other hand, the soil column is divided into two layers, which represent respectively the upper soil layer and the deep weathered rocks (SSF-DWF model). These developments enhance the degree of refinement of the soil physics described in the model. "*

**P3, l76: "upper soil moisture hourly measurements" –> what's the meaning of upper in this context?**
At 5cm, 10cm, 20cm and 30cm depth. For clarity purpose, 'upper' is removed

**P3, l77: "kilometric resolution" this is a bit ambiguous, please use same definition of spatial resolution as used in lines 73 and 75.**
"kilometric resolution" is replaced by '1-km' resolution

**- The relationship between the various model components was not directly clear to me. You might consider to add a figure in section 2.1 showing the connections between the model components.**
A figure is added in the 2,1 section to summarise the main state variables and flux regarding soil processes for the three soil models. A figure presenting the connections between the MARINE components can be found in Roux et al, 2011. This sentence is added is section 2.1.1 : *" The connections between the model components are extensively described in Roux et al, 2011 "*

**- Although the structure of the manuscript is good, the authors provide a lot of information on models and datasets. The readability of the manuscript would greatly improve when a figure showing the research methodology and a table summarizing all data and models used in the study would be added.**

In order to clarify the description of the three models, a figure (Figure 1) presenting a scheme for the soil module of each model is added in section 2.1, as well as the following lines :
*" This section presents the base version of the MARINE model as propose by Roux et al. (2011), together with the two evolved versions of the model implemented by*

[Figure]

*Douinot et al. (2018) for soil processes description. The figure 1 summarizes the main state variables and flux regarding soil processes for the three versions of MARINE. "*
In addition, the table 2, presented before, summarizes the data used in the study.

**- P4, l116: What exactly are the hypotheses here? Do you mean the model developments? I would not consider that a hypothesis.**
" hypotheses " is replaced by " developments "

**- P5, l131-135: Do you have any references for these datasets?**
For the IGN 25m DEM : IGN- BD Topo

```
(https://geoservices.ign.fr/ressources_documentaires/Espace_documentaire/
BASES_VECTORIELLES/BDTOPO/DC_BDTOPO_3-0.pdf)\\
```

For the INRA soil data base :Robbez-Masson, J., Barthes, J., LEGROS, J., et al.: Bases de données pédologiques et systèmes d'informations géographiques. L'exemple de la région Languedoc-Roussillon., Forêt méditerranéenne, 2000.
These references are added in the text.

**- P5, l141, p6, l153: please add reference to reference list.**
The references for the soil, land cover and DEM data are given in lines 131-135

**- P5, l143-145: How did you define these depths? Also, what exactly do you mean with a shallow depth (for Orbieu)?**

The soil depth values are taken from the INRA soil database, quoted before. " Shallow soils " stands for soil depths around 1cm (no altered soil). These databases

have been established for agronomic uses and do not document deep‐weath-ered rock horizons (i.e. pedologic horizons of type C and deeper) (Vannier et al., 2014).

**- P6, l152: What is the context of critized in this sentence?**
" critized discharges" are defined as discharge measurements that have been corrected from known biases. In particular, in the case of flood events, high water heights are difficult to extrapolate from the rating curve. Direct measurements have then to be criticized and corrected according to humain expertise. This analysis is carried out by the forecasters of the French national flood forecasting services.

**- P6, l158-159: Can you discuss why the response of discharge to precipitation during this flood was so fast?**
A very specific pattern of precipitation occurred during this event. The precipitation field was oriented along the main axis of the river, resulting in intense and devastating surface runoff. This may explain such a speed for this flood. The recent paper of Caumont et al. (2020) extensively describe this flood event. This reference is added to the text.

**- It was not clear to me why you introduce the various soil moisture products in section 2.3. Please clarify at the start of this section why you need these products.**

The aim of this paper is to assess the performances of three soil representation for the representation according to five products available for soil moisture estimation : the SIM product, the LDAS-Monde product, the CGLS Soil Water Index product and the SMOSMANIA measurement. Therefore, these products are data used for the study. As a consequence, they are presented in the data section. A table is added in section

2.3 to summarize these five soil moisture data source.

**- P7, l174: what is the depth of the root layer in SIM?**
In ISBA-3L (used for the SIM1 version of SIM), the root zone moisture corresponds to the humidity of the second soil layer. This depth is spatially distributed and it is parametrized with the soil maps given as entries of the model. In ISBA-DIF (used for the SIM2 version of SIM), the humidity of the root zone is considered as the sum of the humidities of the ISBA-DIF layers between 10 cm and 30 cm deep.

**- P8, line 184: Is LDAS-Monde a data-assimilation framework? Please make this clear.**
The following sentence is added : *" LDAS-Monde is a data-assimilation framework that assimilates satellite derived data into the ISBA land surface model "*

**- P9, line 214: SMOS remote data –> please clarify that these data are obtained using the SMOS satellite.**
Reformulated as : " from the SMOS satellite data "

**- Figure 2 is not clear to me. The labels are too small and not in English. Please make a clear distinction between SSM and SWI. Also, the difference between 1km and 25km resolution is not clear. In addition, the precipitation data is not visible, consider adding them in a subplot. Furthermore,according to the manuscript, the figure also shows the respective fraction of missing values. I do not see this in the figure.**
The graphical appearance of this figure is enhanced. The figure is sent to the appendix section.

**- P9, l233: How do you define the very local scale?**
Reformulated as : " at the catchment scale ".

**- P11, l256: the authors state that the ADES locations are situated in the study area. However, according to figure 1, the ADES stations are situated outside the catchments. Why do you consider these stations?**
The ADES station mesure the piezometric level of the water tables that infiltrate and exfiltrate within the studied catchments. Despite the ADES station are located outside of the catchment, the associated water tables influence the hydrology of the catchments. The use of the piezometric measurements had been attempted in order to add more validation data source. However, according to reviewer n°1 comments, the reference to the ADES data, as well as the values in the tables and graphics are removed. The ADES network is only quoted in the last paragraph 4.2.3 Water content of the deep layer, as a possible opening for further works.

**- P11, l257-258: What is meant with "the water table is 110 km2 large"? Is this the size of that specific aquifer? If so, what is the relation with the in situ soil moisture station? According to figure 1, the groundwater and soil moisture stations are located quite far away from each other.**

As stated above, the reference to the ADES data, as well as the values in the tables and graphics are removed. The respective sizes of the water tables are provided by the ADES national database. The groundwater and soil moisture stations do not mesure the same variable : the ADES stations mesure deep ground water using piezometers, whereas the SMOSMANIA stations mesure the upper soil water content.

**P11, l265-266: Please describe the eleven soil layers of the LDAS-Monde**

**product in the data section.**
The following sentence is added in section 2.3.2 :
*" For the two considered catchments, the soil column is discritized into 11 layers, with fixed depths. The depth of the total soil column is 300 cm for the two catchments. "*

**- P11, l275: "is compared to the moisture of the surface layer" –> Do you mean compared to the surface layer of LDAS-Monde?**

The synthese variables 'surface layer' and 'deep layer' moisture for the LDA-Monde product are defined line 273. For clarity purpose, the following sentence is added :
*" The moisture of the surface layer is noted $HU_{surf}$ and it is computed as the average of the layer 1 to 5, weighted by their respective depths. The moisture of the deep layer is noted $HU_{deep}$ and it is computed as the average of the layer 6 to 11, also weighted by their respective depths. "*

**- P12, l286-289: I don't understand your argumentation here. Why would you consider the drainage network in averaging of a grid/mesh? Also, do you have 16 different grids for the analysis and you choose to exclude 4 of them? Or are you referring to individual cells of the grid? Could you rephrase and clarify?**

The question is to compare the local SMOSMANIA measurements with the gridded outputs of MARINE. It could be possible to directly extract the grid cell corresponding to the SMOSMANIA point. However, since the LDAS-Monde cells are 1km large, it is more consistent to average the MARINE cells over the same surface. Among the MARINE cell, some are part of the river network (drainage cells). Soil saturation in these cells is very high because the exfiltration law is not the same than is other 'regular' cells in order to better correspond to the exchange between river water and

groundwater. In consequence, the cells corresponding to the river network are not considered in the averaging.

For clarity purpose, L287-289 are reformulated as :
*" In addition, among the MARINE grid cells, some are part of the river drainage network. As the physics of the soil saturation in the drainage network are not the same than over hillslope cells, the cells corresponding to the MARINE drainage network are excluded from the 1 km$^2$ area around the measurement point. For the Ardeche catchment, 4 drainage cells are excluded from the 16 cells around the measurement point. For the Orbieu catchment, no drainage cells are located within 1 km$^2$ around the measurement point, so no cells are excluded. "*

**- P13, l291: Add reference for NS-efficiency criterion. Also, is LNP index an abbreviation? Furthermore, please add the units of each term in equation 1.**
The reference for NSE is added. LNP is the acronyme of " Likelihood using Nash and Peak " but is not quoted as so in the original paper.In equation 1, each term is adimentional. The unit of the discharge and time variables are added in 3.1.3.

**- P13, l317-318: Why do you consider two grids with different spatial resolution? Can you discuss the impact on your results? Also, why do you have a computational time step of 5 minutes while the precipitation input data has a hourly time step? Please explain.**

The Orbieu catchment is significantly smaller than the Ardeche catchement, so the use of a finer resolution is reasonable. In addition, the calibrations of MARINE used in this work have been performed Garambois et al. (2015) for the Orbieu catchment, using the 200 m resolution and by Douinot (2016) for the Ardeche catchment using

the 250 m resolution. The resolutions used for the calibrations have to be kept for the consistency of simulations using these calibrations.

These lines are reformulated as :

*" The model is set up over a regular mesh. The spatial resolutions applied by Garambois et al. (2015) and Douinot et al. (2018) for the calibration are kept. For the Orbieu catchment, the spatial resolution is 200 m and 250 m for the Ardeche catchment. Despite the precipitation information is given at the hourly time step, the sub-hourly processes are simulated using a 5 minutes computation time step and results are aggregated at the hourly time step. "*

**- Section 3.2.2: Shouldn't this section be part of the results section?**

As the calibration of the models are taken from previous study, the discharge simulations are not considered as results from this current work, but rather as part of the methodology applied for model inter-comparison on soil moisture simulation.

**- Table 4: maybe a figure would visualize these data better, or add the numbers in figure 5.**

We found no satisfactory option to plot all this amount of values on a single plot as the figure 5 is already over crowded.

**- P15, l334-336: I don't understand the argumentation here.Is your argument here that you use the same parametrization for the BM, SSF, and SSF-DWF model variants? However, the SSF and SSF-DWF model variants contains more parameters, so how do you cope with that?**

The model set-up used in the publication is based on the calibration of the base model BM. The following sentence is added to the text :
*" As the SSF model doesn't involved additional parameters, the same calibration is used for the SSF and the base model, given by Douinot et al. (2018) for the Ardeche catchment and Garambois et al. (2015) for the Orbieu catchment. The SSF-DWF model involves to also calibrate the depth of the deep layer. Therefore, the calibrations of the SSF-DWF model performed by Douinot et al. (2018) for both the Orbieu and the Ardeche catchment are used."*

**- Although you state otherwise on p19, l405-407, the initialization grid shown in Figure 8 is still visible in the flood rising stages of the SSF and SSF-DWF models. Is the model able to reach an equilibrium after initialization or is the model still in a spin-up stage? Did you had a look at this?**

As the model is event-based, it only runs for time windows of the duration of the flash-flood, that is to say from several hours to a few days. There is no spin-up stage possible for that kind of models and it is therefore likely that the initialization does have an impact on the simulation results. That is why a systematic initialization has been chosen for all the models using the spatial soil saturation outputs from Météo-France's SIM (Safran-Isba-Modcou, Habets et al., 2008) operational chain : the initial soil moisture is not calibrated.

**- P20, l432-433: "The computation of spatial moments for the CGLS SWI might not lead to robust conclusions."  Ok, so what is the consequence for your message? Why even considering these data in the manuscript then?**
As mentioned in the text, the amount of missing pixels is important in the CGLS SWI product, in particular for the Ardeche catchment. In consequence, this data source is not reliable to assess the spatial repartition of the soil moisture.  However, the CGLS SWI remains valuable to assess the dynamics of the catchment average soil moisture. Indeed, the available pixels are informative when the data is averaged over

the catchment.

**- P23, l454: "Additional research regarding the deep layer calibration should be led." Please rephrase and explain why additional research should be performed.** A more robust calibration is needed for the deep layer, ideally based on the knowledge of the characteristics of the flow in the weathered bedrock. A sentence has been added in this direction : *" Additional research regarding the deep layer calibration should be led. In particular, the Height Above Nearest Drainage (HAND) method would offer the opportunity to take into account the terrain physical characteristics in the deep layer parametrization (Nobre et al., 2011). "*

**- Figure 11: I am not convinced that you can use the piezometric observations for validation of the deep layer of the SSF-DWF model. (1) please explain why you validate soil moisture simulations with groundwater observations. Or do you investigate groundwater simulations here?(2) the groundwater observations are located outside the study area. Are they valid to use?**

As mentioned before, despite the ADES station are located outside of the catchment, the associated water tables influence the hydrology of the catchments. The use of the piezometric measurements had been attempted in order to add more validation data source. However, the authors agree that the time scale of the variations of the water table does not allow to use piezometric values to asses the simulations of ground water here. The reference to the ADES data, as well as the values in the tables and graphics are removed. The ADES network is only quoted in the last paragraph 4.2.3 Water content of the deep layer, as a possible opening for further works.

**- Conclusions: please remove references from conclusion section. too much**
**information is provided in the conclusion section. Please make the section more
concise and to the point.**

The summary of the method is removed from the conclusion. The conclusion is
shortened and some parts are reformulated as :

*The local comparison of the MARINE outputs for surface soil saturation with the
SMOSMANIA measurements, as well as the comparison at the basin scale with the
gridded LDAS-Monde and CGLS data lead to the same conclusions: SSD simulated
with the base model significantly differs from the simulations using the SSF and the
SSF-DWF models. When no precipitation happens, the soil layer empties faster with
the base model, leading to a simulated SSD significantly lower with the base model
than with the two other models. This behavior can be physically explained by the fact
that, in the SSF and the SFF-DWF models, the lateral transfers are computed as a
function of the volumetric soil water gradients, whereas in the base model, they are
computed as a function of the water height gradient. Indeed, since the water height
gradient between two cells depends on the slope between the cells and the cells
textures, water height gradients are larger than volumetric soil water gradient when no
precipitation happens. Consequently, lateral flows based on the water height gradients
are larger than lateral flows based on the volumetric soil water gradient. In addition,
the dynamics as well as the amplitudes of the SSD simulated in the SSF model
and for the upper layer in the SSF-DWF model are better correlated with both the
SMOSMANIA measurements and the LDAS-Monde data than the outputs of the base
model. Considering that the dynamics of the LDAS-Monde $HU_{surf}$ are of satisfying
accuracy, this assessment leads to the conclusion that the SSF-DWF model improves
the simulation of the dynamics of the surface layer moisture, compared to both the
SSF and the base models. This results appears to be particularly reliable, since it is
observed both a the point measurement scale and at the catchment scale.*

*In the SSF-DWF model, the simulation of the moisture in the deep layer is also
compared to LDAS-Monde moisture data provided for deeper layers. However, the*

*simulation of the deep layer water content strongly depends on the calibration of the deep layer thickness, the deep layer porosity and the vertical and lateral hydraulic conductivities in the deep layer. These results illustrate the difficulty to represent the hydrological dynamics of the deep soil layers, with limitation due to the lack of knowledge concerning the physical description of the subsurface water storage. Further conclusions concerning the simulation of deep SSD would then require an extensive work to enhance the parametrization of the deep layer in the SSF-DWF model.*

*In conclusion, this work exposes that computing the infiltration flow as a function of the soil saturation degree instead of the water height in the MARINE model enhance the soil moisture simulation during flash floods, with respect to both local measurements and spatially distributed products.*

**- P27, l529: the HAND method is introduced in the conclusions, but not discussed in the rest of the manuscript. Either discuss the HAND method in the discussion or remove this line from the conclusions.**

The HAND method is quoted in the 4.2.2 Result section :
*" Additional research regarding the deep layer calibration should be led. In particular, the Height Above Nearest Drainage (HAND) method would offer the opportunity to take into account the terrain physical characteristics in the deep layer parametrization (Nobre et al., 2011). "*

**- P27, l531-534: "In conclusion, this work exposes that enhancing the degree of refinement of the soil physics for the representation of subsurface flow in the**

MARINE model appears to enhance the upper soil moisture simulation during flash floods, with respect to both spatialized model outputs and satellite-based data, as well as with respect to local soil moisture measurements." –> This final statement is difficult to read and follow, although I believe this sentence is strong in summarizing the entire manuscript. The authors should clearly rephrase this sentence.

This sentence is reformulated as : " *In conclusion, this work exposes that computing the subsurface flow as a function of the soil saturation degree instead of the water height in the MARINE model enhance the soil moisture simulation during flash floods, with respect to both local measurements and spatially distributed products.* "

**- The authors refer to several articles written in French, such as PhD theses. Is the work of these theses not published in English journal articles or other works written in English?**

When possible, the references to PhD are replaced by the reference to the corresponding paper.

**Technical comments:**

All the technical comments and spelling errors have been corrected in the text.

**References**

Brocca, L., Melone, F., Moramarco, T. and Singh, V. P., 2009. Assimilation of Observed Soil Moisture Data in Storm Rainfall-Runoff Modeling. Journal of Hydrologic Engineering, 14(2). doi: 10.1061/(ASCE)1084-0699(2009)143A2(153).

Caumont, O., Mandement, M., Bouttier, F., Eeckman, J., Lebeaupin Brossier, C., Lovat, A., Nuissier, O., and Laurantin, O.: The heavy precipitation event of 14–15 October 2018 in the Aude catchment: A meteorological study based on operational numerical weather prediction systems and standard and personal observations, Nat. Hazards Earth Syst. Sci. Discuss., https://doi.org/10.5194/nhess-2020-310, in review, 2020.

Habets, F., Boone, A., Champeaux, J.-L., Etchevers, P., Franchisteguy, L., Leblois, E., Ledoux, E., Le Moigne, P., Martin, E., Morel, S., Noilhan, J., Quintana Seguí, P., Rousset-Regimbeau, F., and Viennot, P.: The SAFRAN-ISBA-MODCOU hydrometeorological model applied over France, J. Geophys. Res.-Atmos., 113, D6, https://doi.org/10.1029/2007JD008548, 2008.

Tramblay, Y., Bouvier, C., Martin, C., Didon-Lescot, J.-F., Todorovik, D. and Domergue, J.-M., 2010. Assessment of initial soil moisture conditions for event-based rainfall–runoff modelling. Journal of Hydrology, 387(3–4), 176-187. https://doi.org/10.1016/j.jhydrol.2010.04.006.

Vannier, O., Braud, I. and Anquetin, S. 2014. Regional estimation of catchment-scale soil properties by means of streamflow recession analysis for use in distributed hydrological models. Hydrological Processes, 28(26). https://doi.org/10.1002/hyp.10101.

---

## Referee Report (RR1)

The authors greatly improved the manuscript and provided acceptable answers to the questions posed in the review. Also, the grammar and spelling are greatly improved. However, some sections need technical corrections, after which I advise to publish the revised manuscript. Some examples:

P2, l26: represent valuable tools -> are valuable tools

P2, l34: rainfall intensity excess --> rainfall intensity exceeds

P2, l53: soil and deep ground description

Title of appendix A: litterature --> literature